# A septo-hypothalamic-medullary circuit directs stress-induced analgesia

Devanshi Piyush Shah[1], Pallavi Raj Sharma[2], Rachit Agarwal[2], Arnab Barik[1]*

[1]Centre for Neuroscience, Indian Institute of Science, Bengaluru, India; [2]Department of Bioengineering, Indian Institute of Science, Bengaluru, India

## eLife Assessment

This **important** work explores the modulation of pain by intense stress. The authors employed a series of cutting-edge techniques and provided **convincing** evidence suggesting that the dorsal lateral septum-> lateral hypothalamus-> rostral ventromedial medulla circuit is responsible for mediating stress-induced analgesia. This work will be of interest to neuroscientists interested in the neural circuits of behavior, and scientists interested in stress or pain.

**Abstract** Stress is a potent modulator of pain. Specifically, acute stress due to physical restraint induces stress-induced analgesia (SIA). However, where and how acute stress and pain pathways interface in the brain are poorly understood. Here, we describe how the dorsal lateral septum (dLS), a forebrain limbic nucleus, facilitates SIA through its downstream targets in the lateral hypothalamic area (LHA) of mice. Taking advantage of transsynaptic viral-genetic, optogenetic, and chemogenetic techniques, we show that the dLS→LHA circuitry is sufficient to drive analgesia and is required for SIA. Furthermore, our results reveal that the dLS→LHA pathway is opioid-dependent and modulates pain through the pro-nociceptive neurons in the rostral ventromedial medulla (RVM). Remarkably, we found that the inhibitory dLS neurons are recruited specifically when the mice struggle to escape under restraint and, in turn, inhibit excitatory LHA neurons. As a result, the RVM neurons downstream of LHA are disengaged, thus suppressing nociception. Together, we delineate a poly-synaptic pathway that can transform escape behavior in mice under restraint to acute stress into analgesia.

*For correspondence:
arnabbarik@iisc.ac.in

## Introduction

Descending modulation of pain is the mechanism through which the brain imparts control over somatosensory information processing in the spinal cord. Forebrain and midbrain regions encoding internal states such as stress and hunger can harness the brainstem circuits that project to the spinal cord neurons to alleviate pain (*Tracey and Mantyh, 2007*). Such analgesic mechanisms are critical for survival since they prepare animals and humans to cope with the stressors or enable them to meet their physiological needs on time. Moreover, harnessing the underlying modulatory mechanisms may lead us to novel therapeutic approaches for chronic pain. SIA is one such incidence where an acute stressor, such as physical restraint, can result in analgesia. However, the neural mechanisms that facilitate SIA remain poorly understood.

Experiments involving human subjects in the 1950s and 1960s have shown that direct electrical stimulation of the dLS has anti-nociceptive (*Gol, 1967*; *Olds and Milner, 1954*) effects. In rats, dLS activation suppressed behavioral responses to foot-shocks (*Breglio et al., 1970*) or sustained noxious stimuli such as intraplantar formalin injections (*Abbott and Melzack, 1978*). Furthermore, septum inactivation or lesion renders rats hypersensitive to sensory stimuli, and as a result, they display exaggerated stimulus-driven defensive responses (*Albert and Wong, 1978*; *Köhler, 1976*). dLS neurons

receive nociceptive inputs from the thalamus and somatosensory cortices, as well as anxiogenic information from the hypothalamus (*Chen et al., 2021*; *Menon et al., 2022*). Neurotensin expressing dLS neurons with projections to the LHA mediate acute stress mediated suppression of food consumption (*Azevedo et al., 2020*). Despite these convincing leads indicating the involvement of dLS neurons in the processing of noxious somatosensory stimuli, mechanistic investigation of the role of dLS in nociception remains scarce. Classical studies involving lesions to lateral parts of the septal nucleus resulted in a phenomenon known as septal rage, which leads to heightened defensive responses against non-threatening stimuli, indicative of increased levels of stress and anxiety (*Brady and Nauta, 1953*; *Brady and Nauta, 1955*). Interestingly, dLS is essential to sensing acute stress and instrumental in stress-induced fear and anxiety (*Anthony et al., 2014*; *Rizzi-Wise and Wang, 2021*; *Sheehan et al., 2004*; *Terrill et al., 2018*). This involvement of dLS in acute stress-induced anxiety is reflected in behavioral phenotypes and the elevated blood corticosterone levels (*Singewald et al., 2011*). Taken together, dLS neurons were shown independently to mediate the effects of both nocifensive behaviors and acute stress. However, the role of dLS neurons in pain modulation in the event of an ongoing stressful stimulus remains unexplored. Hence, we hypothesized that dLS might play a facilitatory role in SIA.

The dLS neurons communicate primarily through their inhibitory neurotransmitters and have prominent projections to brain areas involved in nociception and pain processing (*Rizzi-Wise and Wang, 2021*). Of note are the projections to the lateral habenula (LHb) and the LHA (*Dafny et al., 1996*; *Shelton et al., 2012*). The midbrain LHb can modulate the affective aspects of pain via its projections to the brainstem periaqueductal gray (PAG) and dorsal raphe (*Shelton et al., 2012*). LHA neurons have strong connections to the pain-modulatory nuclei in the brainstem, such as the rostral ventromedial medulla (RVM) and the lateral parabrachial nuclei (LPBN). In addition, recent studies have reported that LHA neurons can sense noxious stimuli and suppress pain (*Siemian et al., 2021*; *Carr and Uysal, 1985*), and specifically, the projections to the RVM were shown to have anti-nociceptive effects (*Behbehani et al., 1988*; *Dafny et al., 1996*; *Franco and Prado, 1996*; *Fuchs and Melzack, 1995*; *Holden and Pizzi, 2008*). Notably, LPBN neurons are one of the primary targets of the ascending nociceptive projection neurons from the dorsal horn of the spinal cord (*Todd, 2010*), and are instrumental in determining nociceptive thresholds (*Chiang et al., 2020*) and mediating nocifensive behaviors (*Barik et al., 2018*; *Arthurs et al., 2023*; *Han et al., 2015*). Thus, we reasoned that the dLS→LHA circuitry, through downstream circuits, may mediate SIA.

Here, we have explored the mechanisms through which acute restraint stress (RS)-responsive dLS neurons recruit downstream circuits to provide pain relief. Taking advantage of the anterograde trans-synaptic and retrogradely transporting viral genetic tools for anatomic circuit mapping, we have traced a pathway that originates in the dLS, and routes through LHA while finally terminating onto the spinally-projecting RVM neurons. Optogenetic and chemogenetic manipulations of the activity of each node of this pathway informed us on how they play interdependent roles in transforming restrain-induced stress into the suppression of acute thermal pain. Fiber photometry recordings revealed how the inhibitory dLS neurons may suppress the excitatory LHA neurons upon stress, which can disengage pro-nociceptive RVM cells, resulting in analgesia. Taken together, our data propose a mechanism that can explain how RS can suppress pain and lead to understanding how the circuitries dedicated to detecting stress can interface with the ones built for modulating pain.

## Results

### dLS neurons drive acute stress and SIA

The dLS neurons have been known to be engaged by stressful stimuli such as physical restraint (*Azevedo et al., 2020*). To confirm this, we compared the expression of c-Fos, a molecular proxy for neural activation, in the dLS of control mice with the mice subjected to the RS assay (see methods). Mice that underwent RS showed increased cFos expression in the dLS as compared to the control unstressed mice (*Figure 1A-C*, *Figure 1—figure supplement 1A*). This suggests that dLS neurons may be involved in mediating acute stress, in agreement with previous studies (*Azevedo et al., 2020*; *Kubo et al., 2002*). Notably, internal states, such as stress, can determine nociceptive thresholds (*Amit and Galina, 1986*; *Butler and Finn, 2009*). Despite extensive investigation into the dLS's role in stress, not much is known about the involvement of these neurons in pain modulation. Thus, we tested

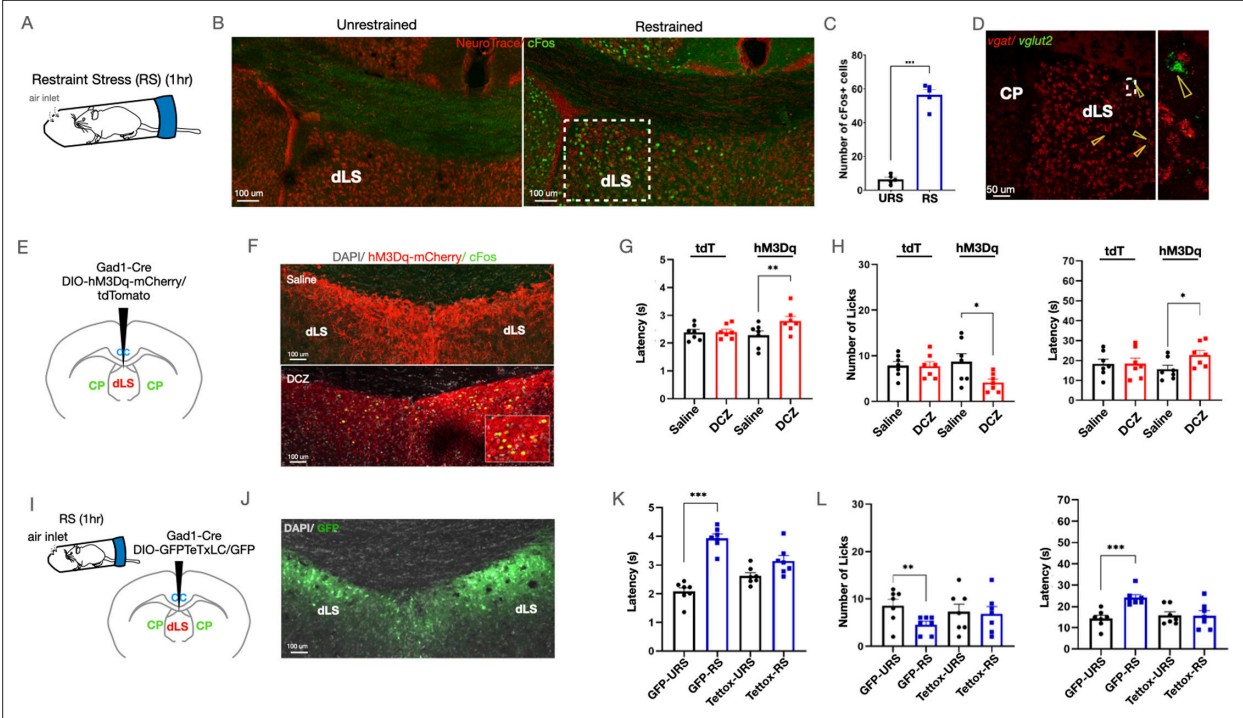

**Figure 1.** Dorsal lateral septum (dLS) neurons are both sufficient and necessary for causing acute restraint-induced analgesia. (**A**) Schematic representing the restraint stress (RS) assay used to induce stress in mice. (**B**) cFos expression in the dLS of restrained mice compared to unrestrained control mice (Green-cFos, Red-neurotrace (nuclear dye specific to neuronal nuclei)). (**C**) Total number of cFos + cells in the dLS (area marked by dotted lines) in restrained vs unrestrained mice (6.40 ± 1.21 compared to 56.40 ± 3.09, respectively; t-test, ***p=0.0001, n=5). (**D**) Multiplex In Situ hybridization with *vglut2* and *vgat* (red-*vgat*, green-*vglut2*), highlighted by arrowheads (left), with a zoom-in view of a rare *vglut2+* (green) cell (right). (**E**) Gad1-Cre and DIO-hM3Dq-mCherry/DIO-tdTomato were injected in the dLS of wild-type mice. (**F**) i.p. Deschloroclozapine (DCZ) and not Saline evoked cFos expression (green) in the mCherry-positive cells (red) in the dLS$^{Gad1-hM3Dq}$ neurons. (**G**) Tail-flick latency (seconds) (2.28 ± 0.16 compared to 2.8 ± 0.16, respectively; t-test, **p=0.0056, n=7) post-saline or DCZ administration in dLS$^{Gad1-hM3Dq}$ mice, with no significant difference in dLS$^{Gad1-tdTomato}$ mice. (**H**) Licks (8.71 ± 1.76 compared to 4.14 ± 0.74, respectively; t-test, *p=0.0335, n=7), and latency to lick (seconds) (15.57 ± 2.03 compared to 22.86 ± 2.26, respectively; t-test, *p=0.0338, n=7) on the hot-plate test of dLS$^{Gad1-tdTomato}$ mice, administered with either i.p. Saline or DCZ, with no significant difference in dLS$^{Gad67-tdTomato}$ mice. (**I**) Gad1-Cre and DIO-Tettox-GFP or DIO-GFP were co-injected in the dLS of wild-type mice. (**J**) GFP-positive neurons (green) are seen in the dLS. (**K**) Tail-flick latency (seconds) (2.08 ± 0.14 compared to 3.9 ± 0.15, respectively; t-test, ***p=0.0001, n=8) with and without restraint for 1 hr using the RS assay in dLS$^{Gad1-GFP}$ mice, with no significant difference in dLS$^{Gad1-Tettox-GFP}$ mice. (**L**) Number of licks (8.57 ± 1.34 compared to 4.57 ± 0.69, respectively; t-test, **p=0.0089, n=7), and latency to lick (seconds) (14.29 ± 1.54 compared to 24.14 ± 1.40, respectively; t-test, ***p=0.0001, n=7) with and without restraint in dLS$^{Gad1-GFP}$ mice, with no significant difference in dLS$^{Gad1-Tettox-GFP}$ mice.

The online version of this article includes the following figure supplement(s) for figure 1:

**Figure supplement 1.** The LS is engaged by acute RS and is involved in SIA.

**Figure supplement 2.** dLS mediated SIA is short lasting as in RS.

if artificial stimulation of the dLS neurons in the CD1 strain of mice can affect the animal's latency to response on the tail-flick test and nocifensive behaviors such as licks on the hot-plate test. Tail-flick and hot-plate tests can reveal the involvement of spinal and supraspinal circuitries in behavioral responses to noxious thermal stimuli, respectively. Since most of the dLS neurons are inhibitory (**Figure 1D**), we expressed the Gq-coupled excitatory Designer Receptors Exclusively Activated by Designer Drugs (DREADD) hM3Dq (**Alexander et al., 2009**) under the inhibitory neuron-specific *Gad1* promoter (dLS$^{Gad1-hM3Dq-mCherry}$) (**Figure 1E and F**). Upon intraperitoneal (i.p.) administration of the ligands CNO or DCZ, hM3Dq enables neuronal firing. When dLS neurons were activated by administration of i.p. DCZ (**Alexander et al., 2009**; **Nagai et al., 2020**), the mice exhibited stress and stress-induced anxiety-like behaviors as confirmed by blood corticosterone levels (**Figure 1—figure supplement 1B**), the light-dark box test (**Figure 1—figure supplement 1C, D**), and the open field test (**Figure 1—figure supplement 1E, F**). Nocifensive behaviors like licks on the hotplate were suppressed post-DCZ administration in dLS$^{Gad1-hM3Dq-mCherry}$ mice, with no changes in the jumping behavior (**Figure 1—figure**

*supplement 1H, I*). The tail-flick latency in these mice was significantly increased, the number of licks on the hotplate was reduced and the latency to lick initiation increased (*Figure 1H, I*). Mice with activated dLS neurons showed higher thresholds for mechanical pain (*Figure 1—figure supplement 1G*). Similar changes in mice behavior were observed when these tests were combined with optogenetic activation of the dLS neurons (dLS^Gad1-ChR2-GFP) (*Figure 1—figure supplement 1K–M*). Furthermore, it was seen that the analgesic effects of dLS neurons lasted for up to 30 min post optogenetic activation (*Figure 1—figure supplement 2A*), and this timeline was like the analgesic effects induced by RS (*Figure 1—figure supplement 2B*). Concluding, that the RS-induced thermal analgesia and pain-modulation attained by artificial activation of dLS neurons are comparable. This phenomenon, known as SIA, has previously not been associated with the septal neurons. Furthermore, to investigate whether these dLS neurons are necessary for bringing about SIA, we blocked the spike-evoked neurotransmitter release of dLS neurons by expressing the tetanus toxin light chain protein fused with GFP (TetTox-GFP) (dLS^Gad1-TetTox-GFP) (*Campos et al., 2018*; *Xu et al., 2012*; *Figure 1I and J*). The efficacy of the tetanus toxin virus was confirmed by testing its effect on mouse behavior in pairing with the excitatory DREADD hM3Dq (*Figure 1—figure supplement 2C, D*). Post-silencing of the dLS neurons, we observed that the stress caused by RS was unable to cause analgesia that was previously seen on both the tail-flick assay and the hotplate test (*Figure 1K and L*). Again, no difference was seen

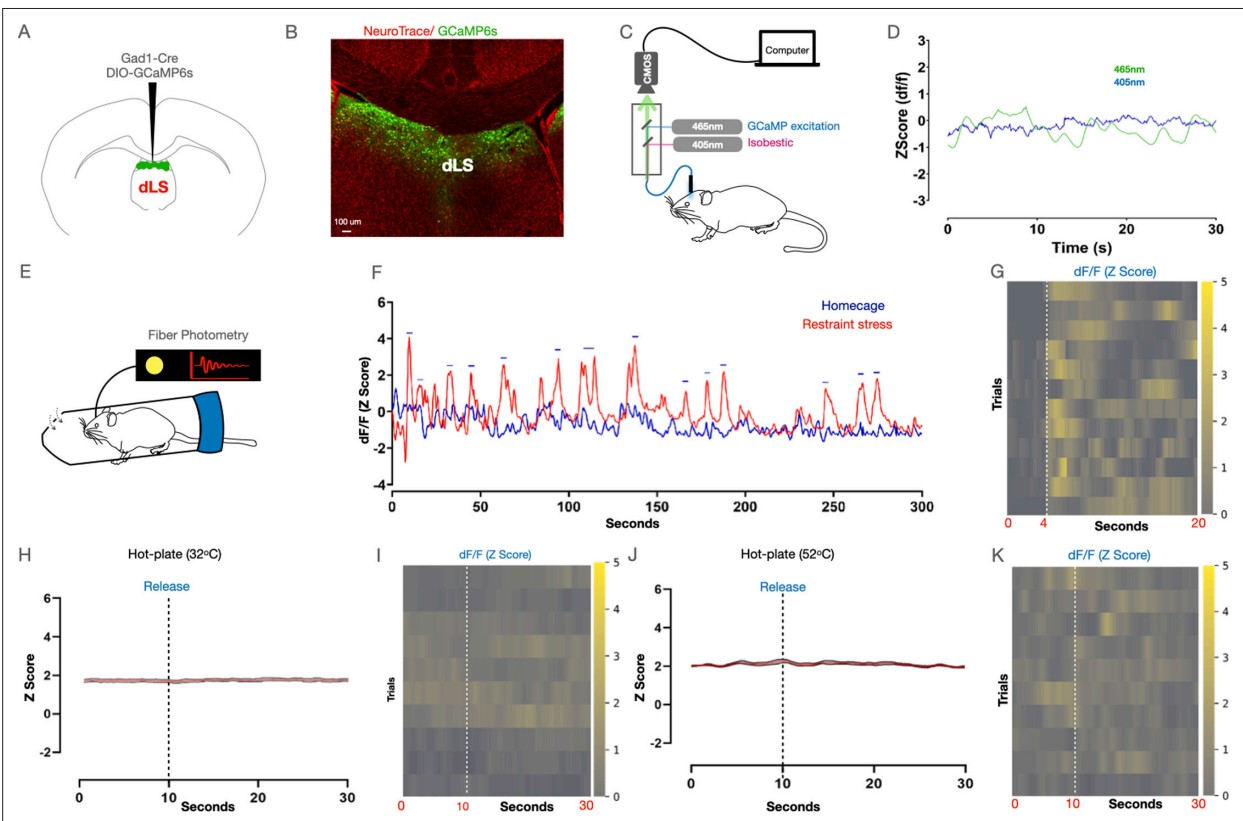

**Figure 2.** Dorsal lateral septum (dLS) neurons are engaged by acute restraint. (**A**) AAVs encoding Gad1-Cre and DIO-GCaMP6s were co-injected in the dLS, and fiber optic cannulaes were implanted above for recording neural activity. (**B**) Confirmation of the expression of GCaMP6s (Green) in the dLS neurons. (**C**) Schematic representation of the fiber photometry system. (**D**) Representative trace or neural activity from GCaMP6s (green) vs control GFP (blue) mice in the homecage. (**E**) Neural activity from the dLS while mice were under restraint. (**F**) Traces of neural activity when mice were allowed to move freely in the homecage (blue), and when they were under restraint (red). Peaks corresponding to neural activity (blue dashes) were seen when mice struggled in the tube. (**G**) Heatmap depicting neural activity during individual instances of struggles (initiation of struggle indicated by white dotted line). (**H and I**) Average plots and heatmaps for dLS responses on the hot plate at 32 degrees (5 mice, 20 trials; dotted line indicating time point when mice were placed on the hotplate). (**J and K**) Average plots and heatmaps for dLS engagement on the hot plate at a noxious temperature of 52 degrees (5 mice, 20 trials; dotted line indicating time point when mice were placed on the hotplate).

The online version of this article includes the following figure supplement(s) for figure 2:

**Figure supplement 1.** Mild stressors and not noxious thermal stimuli engages dLS neurons.

in the jumping behavior of the mice (*Figure 1—figure supplement 2E–G*). Concluding, that the dLS neurons are both sufficient and necessary for SIA.

Next, we probed if the dLS neurons are engaged by RS and/ or noxious stimuli. To that end, we expressed the genetically encoded calcium sensor GCaMP6s (*Chen et al., 2013*) in the dLS neurons (dLS$^{Gad1-GCaMP6s}$) (*Figure 2A and B*), and performed fiber photometry recordings (*Figure 2C*). Fiber photometry enables activity monitoring from the genetically and anatomically defined neuronal population in behaving mice and can provide critical insights into how neural activity corresponds to animal behavior in response to a particular stimulus (*Lerner et al., 2015*). As expected, we observed spontaneous transients in the dLS neurons of mice expressing GCaMP6s but not in the neurons with GFP (*Figure 2D*). Notably, in mice under RS, dLS$^{Gad1-GCaMP6s}$ neurons were active only when the animals in the RS assay apparatus (see methods) struggled and not when they were freely moving in their home cages (*Figure 2E, F, G*, *Figure 2—figure supplement 1B*). This suggests that the increased neural activity during the struggle bouts in mice under RS is not simply due to the increased physical activity but due to the need and inability to escape the restraint (*Figure 2—figure supplement 1A, B*). Similarly, we restrained the mice acutely by hand, and dLS$^{Gad1-GCaMP6s}$ neurons were active during the initial immobilization phase while the mice struggled, after which the activity was reduced when they were unable to move due to the restraint (*Figure 2—figure supplement 1C, D*). The neurons were active again when the mice were released from the hand restraint and were free to run away (*Figure 2—figure supplement 1C, D*). Thus, in agreement with the previous result, the dLS$^{Gad1-GCaMP6s}$ neurons were active when the mice actively struggled to escape any physical restraint. Another acute stress assay that engaged the dLS$^{Gad1-GCaMP6s}$ neurons was the tail hanging assay, wherein mice were suspended in the air by holding their tails for 10 s (*Figure 2—figure supplement 1E, F*). Since artificial stimulation or blocking dLS$^{Gad1}$ neuronal activity had suppressive effects on pain thresholds, we next tested if the dLS$^{Gad1-GCaMP6s}$ neurons respond to noxious thermal stimuli (52 degrees for 1 min on the hot plate). The dLS$^{Gad1-GCaMP6s}$ neurons were found to be inactive during the hot-plate test at innocuous (*Figure 2H and I*), as well as at a noxious temperature when the mice exhibited nocifensive behaviors such as licks and shakes (*Figure 2J and K*). Thus, from our fiber photometry data, we concluded that the dLS$^{Gad1-GCaMP6s}$ neurons are preferentially tuned to stress caused by physical restraint and not to noxious thermal stimuli.

## dLS neurons facilitate SIA through downstream LHA neurons

In the following experiments, we sought to determine the postsynaptic targets through which dLS neurons signal and facilitate RS-induced analgesia. To address this, we mapped the axonal targets of dLSGad1 neurons by labeling these neurons with cell-filing GFP (dLS$^{Gad1-GFP}$) (*Figure 3A and B*). In agreement with previous reports, we observed dLS$^{Gad1-GFP}$ axon terminals in the LHA, the habenula (Hb), and the hippocampus (*Figure 3C*; *Amit and Galina, 1986*; *Kubo et al., 2002*). To confirm that the axonal terminals of the dLS neurons at the targets form synapses with the neurons in the LHA, Hb, and hippocampal formation we labeled the axon terminals of dLS$^{Gad1}$ neurons with synaptophysin fused with the red fluorescent protein ruby (dLS$^{Gad1-SynRuby}$) (*Figure 3—figure supplement 1A*) and found SynRuby puncta in all aforementioned dLS$^{Gad1}$ axonal targets (*Figure 3—figure supplement 1B*), suggesting that dLS neurons synapse onto its downstream LHA (*Figure 3—figure supplement 1C*), Hb, and hippocampal neurons. We hypothesized that the dLS neurons may exert their analgesic effects through downstream LHA and/ or Hb connections for the following reasons: firstly, both LHA and Hb neurons have been implicated in determining nociceptive thresholds (*Singewald et al., 2011*; *Butler and Finn, 2009*); second, LHA and Hb have direct connections to pain modulatory regions in the brain stem such as the PAG, the RVM, and the dorsal raphe nucleus (DRN) (*Singewald et al., 2011*; *Ma et al., 1992*); third, LHA and Hb contain pro-nociceptive neurons, inhibition of which can cause analgesia (*Benabid and Jeaugey, 1989*; *Mahieux and Benabid, 1987*) and fourth, the role of hippocampal neurons in pain modulation is less clear.

To test if dLS modulates nociceptive thresholds through its projections to the LHA, we selectively stimulated the axon terminals of the dLSGad1 neurons in the LHA by expressing ChR2 in the dLS (dLS$^{Gad1-ChR2}$) and shining blue light on the terminals at LHA through fiber optic cannulae (*Figure 3D*). Selective activation of the dLS$^{Gad1-ChR2}$ terminals in the LHA resulted in increased latency on the tail-flick assay (*Figure 3E*) and a reduced number of licks with an increased threshold on the hot-plate assay (*Figure 3F*), suggestive of analgesia. These observations were like the results from the experiments

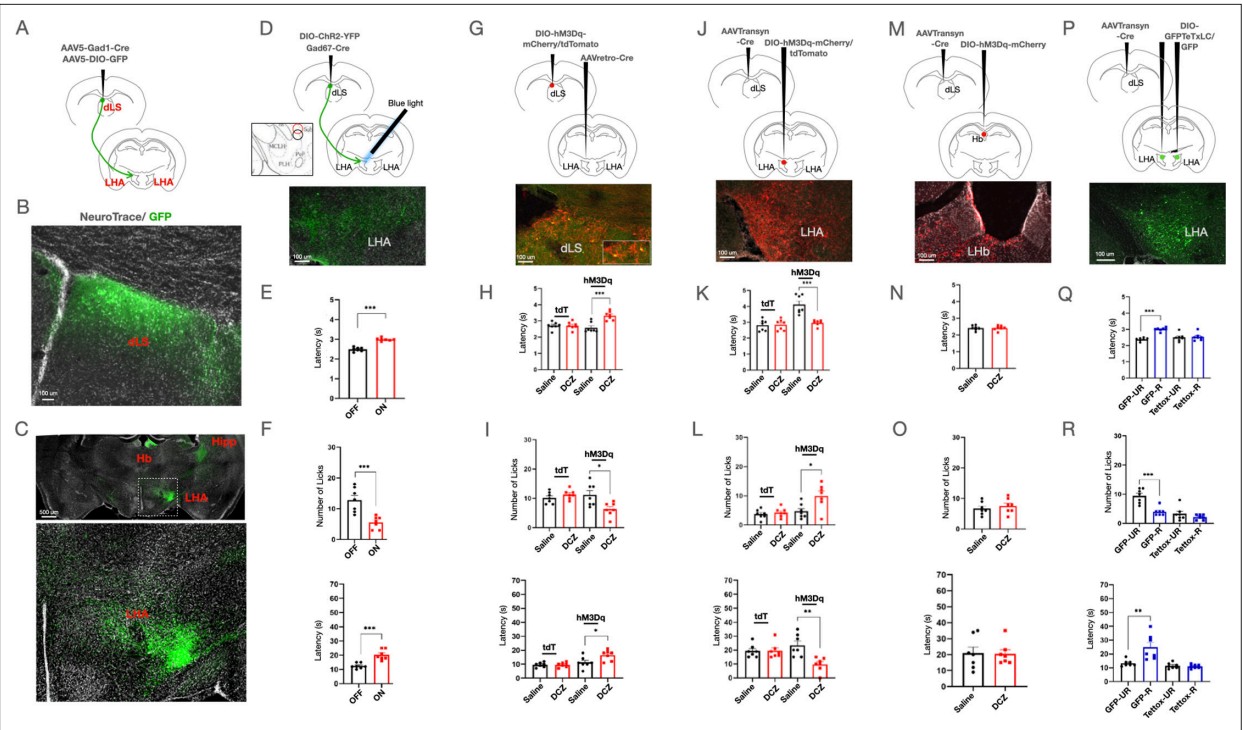

**Figure 3.** Dorsal lateral septum (dLS) neurons synapse onto excitatory lateral Hypothalamus (LHA) neurons. (**A**) Gad1-Cre and DIO-GFP were injected in the dLS of wild-type mice. (**B**) GFP-positive cells seen in the dLS. (**C**) GFP-positive terminals from the dLS were seen in the Hippocampus (Hipp), Habenula (Hb), and Lateral Hypothalamus (LHA), marked region on LHA zoomed-in on the bottom. (**D**) Gad1-Cre and DIO-ChR2-YFP were injected in the dLS of wild-type mice to express the excitatory opsin ChR2 in the dLS$^{Gad1}$ neurons. The fiber was implanted in the LHA to facilitate terminal activation. ChR2-YFP-positive terminals were observed in the LHA. (**E**) Tail-flick latency (seconds) (2.49 ± 0.04 compared to 2.30 ± 0.03, respectively; t-test, ***p=0.0001, n=7) with (ON) and without (OFF) blue light illumination in dLS$^{Gad1-ChR2}$ mice (**F**) Licks (12.86 ± 1.50 compared to 5.57 ± 0.75, respectively; t-test, ***p=0.001, n=7), and latency to lick (seconds) (12.43±0.72 compared to 22.29±1.34, respectively; t-test, ***p=0.0002, n=7) on the hot plate with (ON) and without (OFF) blue light illumination in dLSGad1-ChR2 mice. (**G**) AAVRetro-Cre was injected in the LHA and DIO-hM3Dq-mCherry/DIO-tdTomato in the dLS of wild-type mice. Presence of mCherry-positive cells (red) co-localized with cFos-positive cells (green) in the dLS (overlap between red and green cells shown in zoom-in box). (**H**) Tail-flick latency (seconds) (2.58 ± 0.12 compared to 3.32 ± 0.10, respectively; t-test, ***p=0.0005, n=7) post saline or deschloroclozapine (DCZ) administration in dLS$_{pre-LHA-Gad1-hM3Dq}$ mice, with no significant difference seen in dLS$_{pre-LHA-Gad1-tdTomato}$ mice. (**I**) Number of licks (11.14 ± 1.45 compared to 6.29 ± 0.89, respectively; t-test, *p=0.0147, n=7), and latency to lick (seconds) (11.29 ± 1.51 compared to 16.43 ± 1.39, respectively; t-test, *p=0.0277, n=7) post saline or DCZ administration in dLS$_{pre-LHA-Gad1-hM3Dq}$ mice. (**J**) AAVTransyn-Cre was injected in the dLS and DIO-hM3Dq-mCherry/DIO-tdTomato in the LHA of wild-type mice. Presence of mCherry-positive cells (red) co-localized with cFos-positive cells (green) in the LHA. (**K**) Tail-flick latency (seconds) (2.97 ± 0.07 compared to 4.12 ± 0.22, respectively; t-test, ***p=0.0003, n=7) post saline or DCZ administration in LHA$_{post-dLS-hM3Dq}$ mice, with no significant difference seen in dLS$_{post-dLS-Gad1-tdTomato}$ mice. (**L**) Number of licks (4.71 ± 0.94 compared to 10.00 ± 1.60, respectively; t-test, *p=0.0149, n=7), and latency to lick (seconds) (23.00 ± 3.11 compared to 9.43 ± 1.93, respectively; t-test, **p=0.003, n=7) post saline or DCZ administration LHA$_{post-dLS-tdTomato}$ mice, with no significant difference seen in dLS$_{post-dLS-Gad1-tdTomato}$ mice. (**M**) AAVTransyn-Cre injected were in the dLS and DIO-hM3Dq-mCherry in the Habenula (Hb) of wild-type mice to express the excitatory Designer Receptors Exclusively Activated by Designer Drugs (DREADD). Presence of mCherry-positive cells (red) in the Hb. (**N**) Tail-flick latency post saline or DCZ administration in Hb$_{post-dLS-hM3Dq}$ mice. (**O**) Number and latency of licks (in seconds) post saline or DCZ administration in Hb$_{post-dLS-hM3Dq}$ mice. (**P**) AAVTransyn-Cre were injected in the dLS and DIO-Tettox-GFP/DIO-GFP bilaterally in the LH of wild-type mice. Presence of GFP-positive cells (green) in the LHA. (**Q**) Tail-flick latency (seconds) (2.38 ± 0.04 compared to 3.00 ± 0.04, respectively; t-test, ***p=0.0001, n=7) with and without restraint in LHA$_{post-dLS-GFP}$ mice, with no significant difference seen in LHA$_{post-dLS-TetTox}$ mice. (**R**) Number of licks (9.43 ± 0.92 compared to 4.00 ± 0.53, respectively; t-test, ***p=0.0003, n=7), and latency to lick (seconds) (13.29 ± 0.89 compared to 24.86 ± 3.47, respectively; t-test, **p=0.0072, n=7) with and without restraint in LHA$_{post-dLS-GFP}$ mice, with no significant difference seen in LHA$_{post-dLS-TetTox}$ mice.

The online version of this article includes the following figure supplement(s) for figure 3:

**Figure supplement 1.** Axonal arborizations of the dLS neurons are found across LHA.

**Figure supplement 2.** CNO delivery via microparticle enables long-term activation of dLS neurons.

where the cell bodies of dLS$^{Gad1-ChR2}$ neurons were stimulated (*Figure 1—figure supplement 1I–L*). Taken together, these results suggest that stimulating dLS$^{Gad1}$ cell bodies or their axon terminals in the LHA is sufficient to cause analgesia. In complementary experiments, we chemogenetically activated dLS$^{Gad1}$ terminals in the LHA. To that end, we devised a novel microparticle (MP)-based delivery system (*Jain, 2000*; *Sharma et al., 2022*) for CNO (for the MP-based delivery system, hM3Dq agonist CNO was used due to greater hydrophobicity, which is essential for MP packaging, compared to DCZ), which can be injected specifically in the LHA where dLS neurons terminate (*Figure 3—figure supplement 2A–C*). Compared to the widely-used method for in-vivo drug delivery at deep brain nuclei through cannulae (*Campbell and Marchant, 2018*; *Stachniak et al., 2014*), our poly-DL-lactic-co-glycolic (PLGA) microparticle (MP-CNO) based system (see methods) can serve as a stable, cost-effective, non-invasive, site-specific, and sustained method for CNO delivery. To demonstrate the efficacy of the MP-CNO, we stereotaxically injected MP-CNO in the dLS of dLS$^{Gad1-hM3Dq}$ and recorded cFos expression in the dLS to confirm that the beads were able to activate the neurons. We observed peak cFos expression on Day 4 (*Figure 3—figure supplement 2D*), which is in agreement with our observations of analgesic behavior in the mice. In the dLS$^{Gad1-hM3Dq}$ mice, MP-CNO was analgesic, and the effects were pronounced between 48–96 hr of delivery (*Figure 3—figure supplement 2E, F*). Remarkably, the MP-CNO-induced analgesia was comparable to that of i.p. CNO administration (*Figure 3—figure supplement 2H, I*). The control mice (dLS$^{Gad1-tdTomato}$) did not demonstrate any difference in behavior over days (*Figure 3—figure supplement 2G–I*). Finally, we delivered MP-CNO in the LHA of dLS$^{Gad1-hM3Dq}$ mice for terminal activation and observed increased tail-flick thresholds, fewer licks, and higher lick thresholds (*Figure 3—figure supplement 2J–L*). Thus, our acute optogenetic and chronic chemogenetic axon terminal activation experiments demonstrate that dLS neurons could bring about analgesia through their projections to the LHA.

Next, we chemogenetically stimulated the dLS neurons whose axon terminals arborize and synapse onto the LHA neurons (dLS$_{pre-LHA}$). To that end, we injected retrogradely transporting AAV (AAVRetro-Cre) in the LHA and DIO-hM3Dq-mCherry in the dLS. This intersectional genetic strategy facilitated excitatory DREADD expression exclusively in dLS$_{pre-LHA}$ neurons (dLS$_{pre-LHA-hM3Dq-mCherry}$) (*Figure 3G*). i.p administration of DCZ in the dLS $_{pre-LHA-hM3Dq-mCherry}$ mice had similar analgesic effects as seen with dLSGad1activation (*Figure 3H and I*), indicating that activation of a specific subset of dLS neurons (dLS$_{pre-LHA}$) is sufficient for altering thermal nociception in mice. In the following experiments, we asked what the effect of chemogenetic activation of the LHA neurons downstream of dLS (LHA$_{post-dLS}$) on thermal nociceptive thresholds would be. To address this, we chemogenetically activated LHA$_{post-dLS}$ neurons by injecting AAVTrans-Cre in the dLS and AAV-DIO-hM3Dq-mCherry in the LHA (*Figure 3J*). I.p. administration of DCZ in mice expressing hM3Dq in LHA$_{post-dLS}$ neurons reduced the latency to react on the tail-flick assay, increased the number of licks, and decreased the latency to lick on the hot-plate test (*Figure 3K and L*). Intriguingly, our data suggest that when activated, dLSGad1 neurons inhibit LHA neurons to cause analgesia, and so when LHA$_{post-dLS}$ neurons are artificially activated, it has the opposite effect on nocifensive behaviors and leads to hyperalgesia. In addition, given that dLS neurons project to the Hb (*Figure 3C*) and Hb neurons have been shown to have anti-nociceptive effects (*Cohen and Melzack, 1985*), we tested if activating Hb neurons post-synaptic to dLS (Hb$_{post-dLS}$) can alter thermal pain thresholds (*Figure 3M*). DCZ administration had no effect on thermal pain thresholds in Hb$_{post-dLS-hM3Dq\ mice}$ (*Figure 3N and O*). Furthermore, we tested the effects of silencing the LHA$_{post-dLS}$ neurons on RS-induced SIA (*Figure 3P*). TetTox-mediated LHA$_{post-dLS}$ silencing abolished SIA (*Figure 3Q and R*). In summary, the LHA$_{post-dLS}$ neurons are functionally downstream of dLS$^{Gad1}$ neurons, and simultaneous transient activation has identical effects on nociceptive thresholds.

## dLS neurons synapse onto the Vesicular glutamate transporter 2 (Vglut2)-expressing LHA neurons

We then tested if the LHA neurons receiving inputs from dLS are excitatory or inhibitory, given that LHA is composed of both *Slc17A6* (encoding the gene for VGlut2) and Vesicular Gamma-aminobutyric acid (GABA) Transporter (VGat) or *Slc32A1* expressing neurons (*Figure 4—figure supplement 1A, B*), with a relatively smaller population of VGlut2-positive neurons (*Figure 4—figure supplement 1A, B*), which have recently been implicated in pain modulation (*Singewald et al., 2011*). To test if VGlut2 +LHA neurons receive direct inputs from dLS$^{Gad1}$ neurons, we took three complementary approaches. First, we injected AAV1-FlpO with anterograde transsynaptic transmission properties

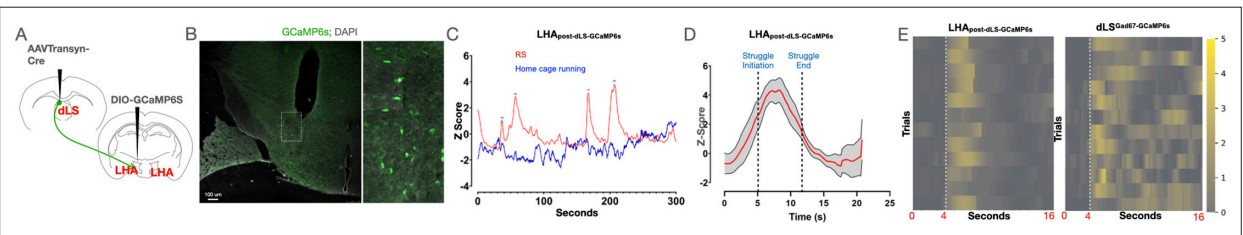

**Figure 4.** Lateral Hypothalamus (LHA)<sub>post-dLS</sub> neurons are glutamatergic. (**A**) AAV encoding AAVTranssyn-FlpO injected in the dorsal lateral septum (dLS); fDIO-tdTomato, and DIO-GFP injected in the LHA of VGlut2-Cre transgenic mice. (**B**) Overlapping red and green cells (yellow, seen in the zoom-in box marked by arrowheads) were seen in the LHA. (**C**) AAVTranssyn-Cre were injected in the dLS of Ai14 transgenic mice. (**D**) The LHA was labeled with probes against *vglut2* using in situ hybridization. Co-localization of tdTomato-positive (red) and GFP-positive (green) cells in the LHA (overlapping areas marked by arrowheads), zoom-in of overlap on right. (**E**) AAVs encoding Gad1-Cre and DIO-Synaptophysin-GFP injected in the dLS and DIO Gephyrin-tagRFP in the LHA of VGlut2-Cre transgenic mice. (**F**) Closely apposed green synaptophysin and red gephyrin puncta were seen in the LHA (zoom-in of overlap on right).

The online version of this article includes the following figure supplement(s) for figure 4:

**Figure supplement 1.** LHA neurons post-synaptic to dLS are excitatory and do not colocalize with PV.

(AAV1-hSyn-FlpO or Transyn-FlpO) (*Alexander et al., 2009*; *Campos et al., 2018*; *Nagai et al., 2020*) in the dLS of VGlut2-Cre transgenic mice (*Figure 4A*). Simultaneously, we injected DIO-GFP and fDIO-tdTomato in the LHA of the same mice (*Figure 4A*). We found that 34.2 ± 9.6% (n=8 sections, 3 mice) tdTomato expressing cells (LHA<sub>post-dLS</sub>) were GFP +ve, indicative of their excitatory status (*Figure 4B*). Second, we delivered the anterograde transsynaptic Cre (AAV1-hSyn-Cre or AAVTranssyn-Cre) in the dLS of Rosa26<sup>LSL-tdTomato</sup> transgenic mice to label the LHA<sub>post-dLS</sub> neurons with tdTomato and performed multiplex fluorescent in-situ hybridization for VGlut2 and tdTomato mRNA in the LHA (*Figure 4C*). We found that 28.5 ± 11.2% (n=12 sections, 2 mice) tdTomato neurons colocalized with VGlut2 (*Figure 4D*). Third, we labeled the synaptic terminals of dLS<sup>Gad1</sup> neurons and post-synaptic densities of LHA-VGlut2 neurons by expressing synaptophysin-fused GFP (SynGFP) in the dLS<sup>Gad1</sup> neurons and Cre-dependent inhibitory postsynaptic protein, Gephyrin fused with red fluorescent protein tagRFP (DIO-GephyrintagRFP) (*Xu et al., 2012*; *Chen et al., 2013*; *Xu et al., 2012*; *Lerner et al., 2015*) in the LHA of VGlut2-Cre mice, respectively (*Figure 4E*). Here, we noticed a close apposition of green synaptophysin and red gephyrin puncta in the LHA (*Figure 4F*), suggesting that dLS<sup>Gad1</sup> neurons make synaptic connections onto VGlut2 neurons in the LHA. Overall, these results suggest that inhibitory dLS axons target the excitatory populations of the LHA neurons. Since parvalbumin (PV) labels a subset of excitatory neurons in the LHA and PV-expressing neurons in the LHA have been shown to be antinociceptive (*Singewald et al., 2011*), we tested if LHA<sub>post-dLS</sub> neurons colocalize with PV-expressing cells and found little to no overlap between the two populations (*Figure 4—figure supplement 1C*).

**Figure 5.** Lateral Hypothalamus (LHA)<sub>post-dLS</sub> neurons are acutely engaged during the initial struggle due to physical restraint. (**A**) AAVTranssyn-Cre was injected in the dLS and DIO-GCaMP6s in the LHA of wild-type mice to record neural activity from the LHA<sub>post-dLS</sub> neurons. (**B**) GCaMP6s-positive cells (green) and tissue injury from the fiber implant seen in the LHA, zoom-in of the marked area on the right. (**C**) Sample trace of neural activity when mice were allowed to move freely in the homecage (blue), and when they were under restraint (red). Peaks corresponding to neural activity (blue dashes) were seen when mice struggled in the tube. (**D**) Cumulative plots for calcium transients when mice struggled under restraint (5 mice, 12 trials). (**E**) Heat maps depicting neural activity in LHA<sub>post-dLS</sub> (left) and dorsal lateral septum (dLS)<sup>Gad1</sup> neurons (right) during struggle in the restraint stress (RS) assay (5 mice, 12 trials; dotted lines indicating initiation of struggle in restraint stress RS assay).

The online version of this article includes the following figure supplement(s) for figure 5:

**Figure supplement 1.** LHA neurons downstream of dLS are activated by acute stressors.

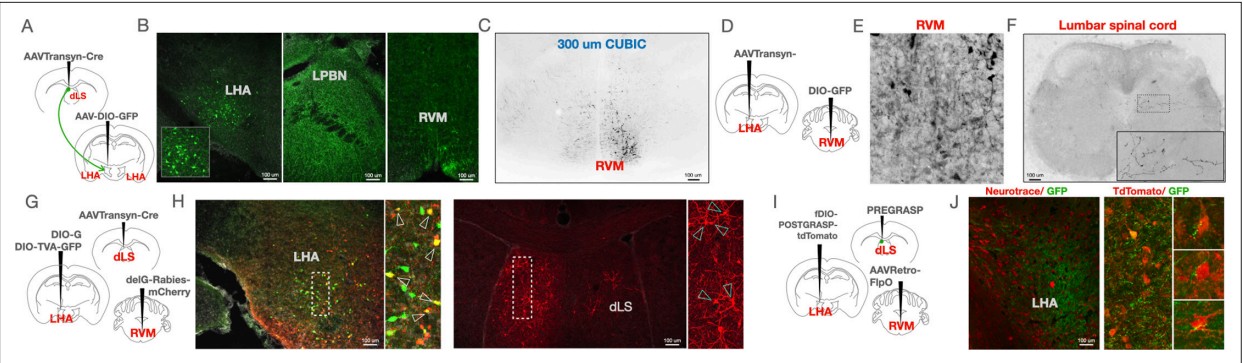

**Figure 6.** Rostral ventromedial medulla (RVM) ON-cells are downstream of the dorsal lateral septum (dLS)→Lateral Hypothalamus (LHA) pathway. (**A**) AAVTransyn-Cre was injected in the dLS and DIO-GFP in the LHA of wild-type mice to label the LHA$_{post-dLS}$ neurons. (**B**) GFP-positive cell bodies in the LHA (green, left; zoom-in of LHA in the box on the bottom). Projections of the LH$_{post-dLS}$ neurons in the PBN, and RVM. (**C**) Presence of fluorescence (black) in RVM post tissue clearing of the same brains expressing GFP in the LHA$_{post-dLS}$ neurons. (**D**) AAVTransyn-Cre was injected in the LHA and DIO-GFP in the RVM of wild-type mice to label the RVM$_{post-LHA}$ neurons with GFP. (**E**) GFP-positive cell bodies (black, marked by arrowheads) are seen in the RVM. (**F**) Projections from the RVM$_{post-LHA}$ were observed in the lumbar spinal cord. (**G**) AAVTransyn-Cre was injected in the dLS, and DIO-ΔG and DIO-TVA-GFP in the LHA of wild-type mice. Three weeks later, the ΔG-Rabies-mCherry virus was injected into the RVM. (**H**) Starter cells (yellow) were observed in the LHA. mCherry-positive cell bodies (red) seen in the dLS (marked region zoomed-in on the right with overlapping cells marked with arrowheads). (**I**) AAVRetro-FlpO injected in the RVM, fDIO-post-GRASP injected in the LHA, and Pre-GRASP in the dLS of wild-type mice. (**J**) Axon terminals from the dLS neurons (Green) were observed around the LHA$_{pre-RVM}$ (red) cell bodies (zoom-in of overlaps on the right).

The online version of this article includes the following figure supplement(s) for figure 6:

**Figure supplement 1.** dLS mediated SIA is opioid dependent.

Together, our anatomical data indicate that the dLS$^{Gad1}$ neurons are synaptically connected with the VGlut2 +ve neurons in the LHA.

## LHA$_{post-dLS}$ neurons are inhibited upon acute restraint

Next, we reasoned that if LHA$_{post-dLS}$ neurons are inhibited by dLS$^{Gad1}$ neurons, then these neurons must be disengaged when mice undergo acute stress. To test this, we labeled LHA$_{post-dLS}$ neurons with GCaMP6s (LHA$_{post-dLS-GCaMP6s}$) (*Figure 5A and B*) and recorded calcium transients from these neurons as mice underwent RS (*Figure 5C*) as well as when they were exposed to noxious thermal stimulus. Surprisingly, fiber photometry recordings showed that LHA$_{post-dLSGCaMP6s}$ neurons respond to acute stress caused by physical restraint and tail hanging (*Figure 5—figure supplement 1A–D*). In addition, like dLS$^{Gad1}$ neurons (*Figure 2H–K*), LHA$_{post-dLS-GCaMP6s}$ neurons were not engaged by noxious thermal stimuli (*Figure 5—figure supplement 1E–H*). Notably, LHA$_{post-dLS-GCaMP6s}$ neurons differed in their activity from the dLS$^{Gad1-GCaMP6s}$ neurons in one aspect — while mice struggled in the RS assay, dLS$^{Gad1-GCaMP6s}$ neurons were active for the entire duration of the struggle bouts (up to 10 s) (*Figure 5D*), LHA$_{post-dLS-GCaMP6s}$ neurons were active only during the initial phase of the struggle (1–3 s) (*Figure 5D*). This indicates that the activity of pre-synaptic dLS and post-synaptic LHA$_{post-dLS}$ neurons increases in a coordinated manner at the onset of struggle in mice undergoing RS-assay. However, after the initial activity, LHA$_{post-dLS}$ neurons can potentially be suppressed by inhibitory dLS inputs causing a reduction in the firing of these post-synaptic neurons. Taken together, these data suggest that acute stress inhibits a sub-population of excitatory neurons in the LHA that are postsynaptic to inhibitory dLS neurons.

## LHA$_{post-dLS}$ neurons facilitate SIA through projections to RVM

Next, in order to investigate how LHA$_{post-dLS}$ neurons facilitate RS-induced analgesia, we mapped the projections of these neurons. Expression of GFP in the LHA$_{post-dLS}$ neurons, labeled axon-terminals in the LPBN, and the RVM (*Figure 6A–C*). In previous studies, activation or silencing of LPBN neurons did not alter the reflexive withdrawal thresholds or coping responses, such as licks in response to noxious thermal stimuli (*Barik et al., 2018*; *Chiang et al., 2020*; *Han et al., 2015*). However, RVM neurons are known to modulate pain bi-directionally (*Fields, 2004*; *François et al., 2017*). Importantly, SIA is opioid-dependent (*Finn, 2017*; *Lewis et al., 1981*; *Vaccarino et al., 1992*; *Figure 6—figure supplement 1A*), and RVM is a major substrate for endogenous opioids (*Fields, 2004*). We confirmed the

opioid-dependent nature of SIA induced by septal activation, by simultaneous administration of mu-opioid receptor antagonist naltrexone and DCZ in dLS$^{Gad1-hM3Dq}$ mice, which lead to the blocking of dLS activation-induced analgesia (*Figure 6—figure supplement 1B, C*). Thus, we reasoned that the projections of the LHA$_{post-dLS}$ neurons synapse onto RVM neurons and facilitate RS-induced analgesia. To that end, first, we sought to establish the anatomical location and projections of the RVM neurons (RVM$_{post-LHA}$) — that are postsynaptic to the LHA$_{post-dLS}$ neurons. We labeled the RVM$_{post-LHA}$ neurons with GFP (*Figure 6D*) and SynRuby (*Figure 6—figure supplement 1D*) separately, using the anterograde intersectional viral genetic strategy used before (AAVTransyn-Cre in LHA; DIO-GFP or DIO-SynRuby in RVM). We observed that the cell bodies of the RVM$_{post-LHA}$ neurons were distributed in the midline area (*Figure 6E*, *Figure 6—figure supplement 1E*) of the medulla. We noticed abundant axon terminals of RVM$_{post-LHA}$ neurons in the LHA (*Figure 6—figure supplement 1F*), and projections in the deeper layers (VI/ VII) of the lumbar spinal cord (*Figure 6E*, *Figure 6—figure supplement 1G*). This implies that the RVM$_{post-LHA}$ neurons may modulate nociceptive thresholds through their local synaptic connections within the RVM (*Marchand and Hagino, 1983*), recurrent connections with the PAG, or direct interactions with spinal cord neurons. Second, using the monosynaptic retrograde rabies tracing technique (*Callaway and Luo, 2015*; *Wickersham et al., 2007*), we determined if the RVM$_{post-LHA}$ neurons are the direct postsynaptic partners of the LHA$_{post-dLS}$ neurons. We expressed G and TVA-GFP proteins in LHA$_{post-dLS}$ neurons (AAVTransyn-Cre in dLS; DIO-G; DIO-TVA-GFP in LHA) and injected ΔG-Rabies-mCherry in the RVM (*Figure 6G*). As expected, we observed the starter cells that co-expressed TVA-GFP (LHA$_{post-dLS}$) and ΔG-Rabies-mCherry (retrogradely transported from the RVM; LHA$_{pre-RVM}$) in the LHA (*Figure 6H*). Remarkably, retrogradely transported rabies-mCherry was found in the dLS neurons (*Figure 6H*). Indicating that the dLS neurons are directly upstream of the RVM projecting LHA neurons. Third, in a complementary approach, we took advantage of the GRASP synaptic labeling strategy (*Kim et al., 2011*), where presynaptic neurons expressed one-half of the GFP protein, and the postsynaptic neurons expressed the other half. At the functional synapses between the two GFP-subunit expressing neurons, GFP is reconstituted and can be

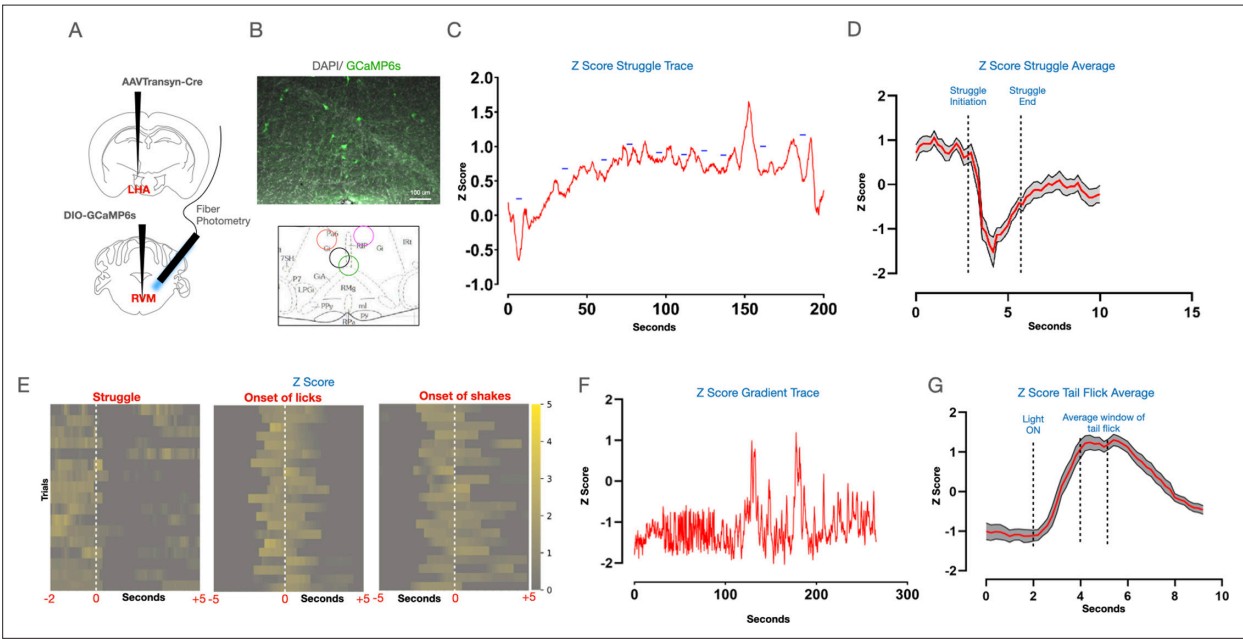

**Figure 7.** Rostral ventromedial medulla (RVM)$_{post-LHA}$ neurons fire during nocifensive behaviors on the hotplate and respond to restraint stress (RS)-mediated struggle. (**A**) AAVTransyn-Cre was injected in the lateral hypothalamus (LHA) with DIO-GCaMP6s in the RVM of wild-type mice (top). A fiber was implanted on the RVM over the same coordinates to perform fiber-photometry (bottom). (**B**) GCaMP6s-positive cells in the RVM (green) to demonstrate successful expression of GCaMP6s in the RVM$_{post-LHA}$ neurons. (**C**) Z-Score of calcium dynamics recorded from the RVM$_{post-LHA}$ neurons while the mice are restrained and struggling (blue dashes) in a falcon tube. (**D**) Average plot (individual trials) of Z-Score traces (4 mice, 16 trials) during struggle bouts of RVM$_{post-LHA}$ neurons. (**E**) (Left to right) Heatmaps depicting neural activity patterns during individual instances of struggle in the falcon tube, licks, and shakes on the hot plate at 52 degrees, respectively (4 mice, 16 trials). (**F**) Z-Score of calcium dynamics recorded from the RVM$_{post-LHA}$ neurons while the mice are on the hotplate set at a gradient from 32 to 52 degrees. (**G**) Average plot (individual trials) of Z-Score traces (4 mice, 16 trials) during tail flick assay of RVM$_{post-LHA}$ neurons.

visualized through a fluorescent microscope. In our experiments, we injected Pre-GRASP in the dLS, fDIO-Post-GRASP in the LHA, and AAVRetro-FlpO in the RVM (*Figure 6I*) of the same mice. Simultaneous injection of the three AAVs successfully labeled the synapses between the dLS and LHA$_{pre-RVM}$ neurons with GFP in the LHA (*Figure 6J*). Thus, demonstrating that the dLS neurons make synaptic connections with LHA$_{pre-RVM}$ neurons.

We then sought to understand how the RVM$_{post-LHA}$ neurons encode RS and noxious thermal stimuli, and thus, we performed fiber photometry recordings from the GCaMP6s expressing RVM$_{post-LHA}$ neurons (RVM$_{post-LHA-GCaMP6s}$) (*Figure 7A and B*). Calcium transients in the RVM$_{post-LHA}$ neurons increased spontaneously when the mice were subjected to RS (*Figure 7C*). However, when the mice struggled under restraint, RVM$_{post-LHA}$ neuronal activity was suppressed (*Figure 7C and D*). The activity of the RVM$_{post-LHA-GCaMP6s}$ neurons increased when the mice shook or licked their paws on the hot-plate test (*Figure 7E*). Notably, the rise in activity of the RVM$_{post-LHA}$ neurons preceded the licks and shakes, indicating a facilitatory role of these neurons in nocifensive behaviors. To further confirm this facilitatory role, performed two additional assays. First, we recorded calcium transients from the RVM$_{post-LHA-GCaMP6s}$ neurons while the mice were on the hot plate set to a gradient of 32–52 degrees over a period of 5 min. We observed that the neurons started firing only once the hot plate reached noxious temperatures, with no specific activity seen at innocuous temperatures (*Figure 7F*). Second, we subjected the mice to the tail-flick assay and observed a peak in neural firing preceding the tail-flick instance caused by the thermal pain caused by the concentrated beam of light (*Figure 7G*). From these series of experiments, we were able to further confirm the facilitatory role of the RVM$_{post-LHA}$ neurons in nocifensive behaviors. The facilitatory pro-nociceptive population of RVM neurons is otherwise known as ON-cells (*Fields et al., 1995*; *Fields et al., 1991*) Thus, we hypothesized that these RVM$_{post-LHA}$ neurons are pro-nociceptive and likely ON cells. They are activated by acute stress and suppressed when the mice struggle to escape stress-causing restraint. When we calculated the area under the curve (AUC) of the Z-score of the recordings from the dLS, LHA$_{post-dLS}$, and RVM$_{post-LHA}$ neurons while the mice struggled in the RS assay, the data indicated that the dLS activity increased while at the same time LHA$_{post-dLS}$, and RVM$_{post-LHA}$ activity decreased (*Figure 8—figure supplement 1E*). Together, our data indicate that the LS inhibitory neurons engaged by acute restraint, suppress LH activity which in turn reduces the excitability of the pronociceptive RVM neurons and result in analgesia.

To further establish that the RVM$_{post-LHA}$ neurons are ON cells, we chemogenetically activated the RVM$_{post-LHA}$ neurons (*Figure 8A and B*). Mice expressing hM3Dq in the RVM$_{post-LHA}$ responded with increased licks on the hot-plate test and a lower latency when i.p. DCZ was administered (*Figure 8C and D*). Contrary to RVM$_{post-LHA}$ activation, when we chemogenetically and silenced the RVM$_{post-LHA}$ neurons (*Figure 8E–H*), the number of licks on the hot-plate test was reduced, and the latency to lick and the tail-flick latency were increased. These findings agree with our hypothesis that the activated dLS neurons inhibit LHA$_{post-dLS}$ neurons, which in turn deactivates RVM$_{post-LHA}$ cells. Moreover, the observation that silencing bi-lateral LPBN neurons postsynaptic to LHA (*Figure 8I and J*) did not affect mouse responses on hot-plate and tail-flick tests confirmed (*Figure 8K and L*) that the LHA-RVM connections primarily mediate the anti-nociceptive effects of LHA$_{post-dLS}$ neurons. The results observed in the experiments with chemogenetic inhibition of RVM$_{post-LHA}$ neurons were repeated with optogenetic inhibition techniques, and similar results were obtained (*Figure 8M–P*).

Recent observations indicate that the pro-nociceptive ON cells in the RVM can be either excitatory or inhibitory (*Nguyen et al., 2022*). Activating the excitatory ON cells results in hypersensitivity to noxious stimuli, whereas inhibiting the same neurons results in analgesia (*Nguyen et al., 2022*). Since the chemogenetic activation of the pro-nociceptive RVM$_{post-LHA}$ neurons resulted in thermal hyperalgesia and inhibition leading to pain suppression, we hypothesized that the RVM$_{post-LHA}$ neurons are excitatory. Thus, to test our hypothesis, we expressed Syn-GFP in the LHA of VGlut2-Cre mice, and PSD95-tagRFP in the RVM of the same mice (*Figure 8—figure supplement 1A–D*). When we visualized the RVM with confocal and super-resolution microscopy (*Figure 8—figure supplement 1D*), we found close apposition between the GFP-expressing pre-synaptic terminals and tagRFP-expressing VGlut2 expressing RVM neurons. Thus, we concluded that the excitatory LHA neurons impinge upon the excitatory RVM-ON neurons to facilitate RS-induced analgesia. The struggle to escape restraint engages dLS inhibitory neurons. The activated inhibitory dLS neurons silence excitatory LHA$_{post-dLS}$ neurons, which consequently disengages the pro-nociceptive RVM$_{post-LHA}$ neurons to drive RS-induced analgesia (*Figure 8—figure supplement 1E–F*).

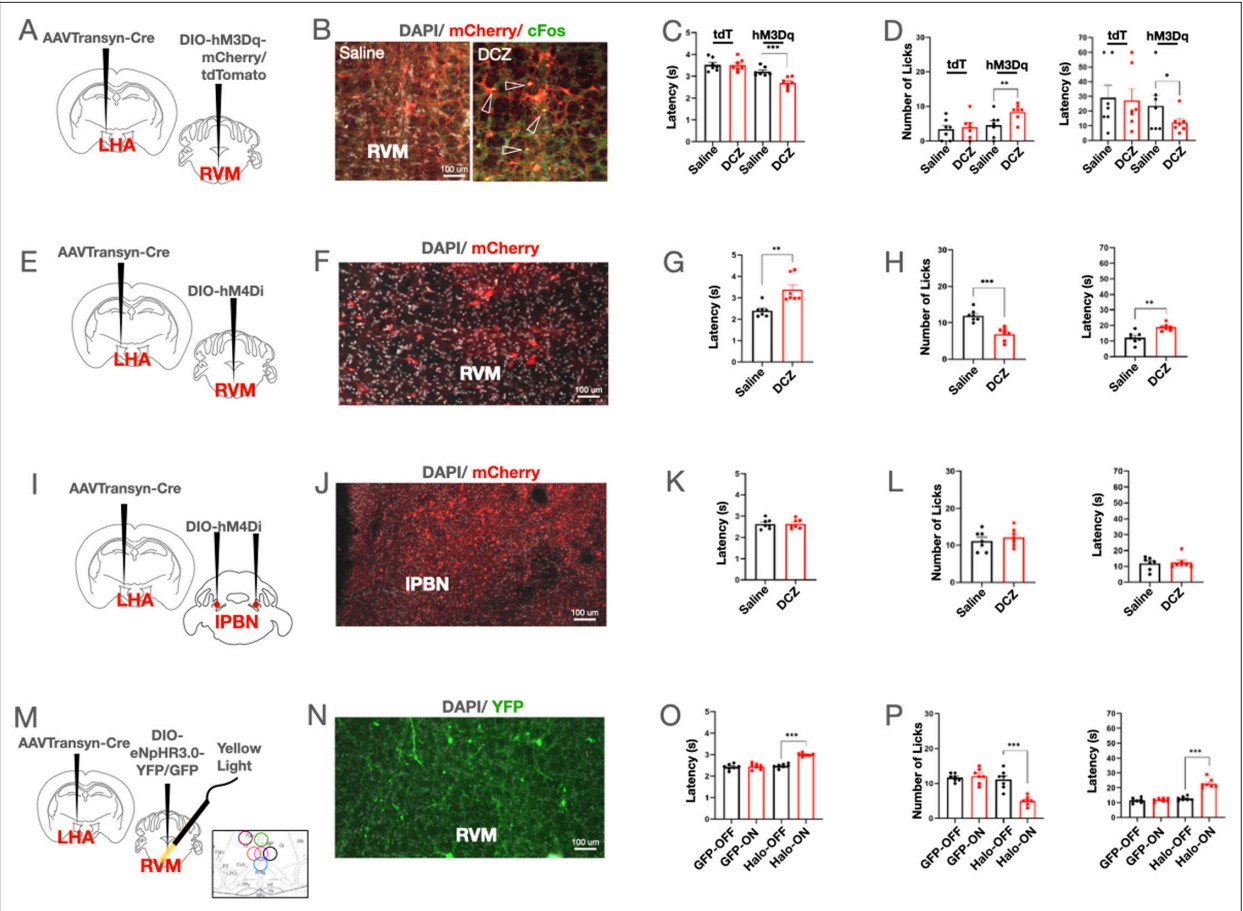

**Figure 8.** Activation of rostral ventromedial medulla (RVM)_post-LHA neurons resulted in thermal hyperalgesia, while inhibition was anti-nociceptive.
(**A**) AAVTransyn-Cre injected in the lateral hypothalamus (LHA) and DIO-hM3Dq-mCherry/DIO-tdTomato in the RVM of wild-type mice.
(**B**) Deschloroclozapine (DCZ) induced c-Fos (green) expression in the RVM_post-LHA-hM3Dq (red) neurons (zoom-in shown on the right with overlapping cells marked using arrowheads). (**C**) Tail-flick latency (seconds) (3.22 ± 0.08 compared to 2.69 ± 0.09, respectively; t-test, ***p=0.0001, n=7) post saline or DCZ administration in the RVM_post-LHA-hM3Dq mice, with no significant difference in RVM_post-LHA-tdTomato mice. (**D**) Number of licks (4.57 ± 1.27 compared to 8.43 ± 0.87, respectively; t-test, **p=0.0062, n=7), and latency to lick (seconds) (23.57 ± 7.12 compared to 12.29 ± 2.68, respectively; t-test, *p=0.0466, n=7) post saline or DCZ administration in the RVM_post-LHA-hM3Dq mice, with no significant difference in RVM_post-LHA-tdTomato mice. (**E**) AAVTransyn-Cre was injected in the LHA and DIO-hM4Di in the RVM of wild-type mice to express the inhibitory Designer Receptors Exclusively Activated by Designer Drugs (DREADD) in the RVM_post-LHA neurons. (**F**) hM3Dq-mCherry expressing neurons (red) in the RVM. (**G**) Tail-flick latency (seconds) (2.40 ± 0.11 compared to 3.38 ± 0.23, respectively; t-test, **p=0.0025, n=7) post saline or DCZ administration in RVM_post-LHA-hM4Di mice. (**H**) Number of licks (12.00 ± 0.62 compared to 6.86 ± 0.67, respectively; t-test, ***p=0.0001, n=7), and latency to lick (seconds) (12.14 ± 1.42 compared to 18.86 ± 0.86, respectively; t-test, **p=0.0016, n=7) post-saline or DCZ administration in RVM_post-LHA-hM4Di mice. (**I**) AAVTransyn-Cre was injected in the LHA and DIO-hM4Di in the PBN (bilaterally) of wild-type mice to express the inhibitory DREADD, hM4Di in the LPBN_post-LHA neurons. (**J**) hM4Di-mCherry (red) expressing neurons in the PBN. (**K**) Tail-flick latency (seconds) post-saline or DCZ administration in the PBN_post-LHA-hM4Di mice. (**L**) Number and latency of licks (seconds) post-saline or DCZ administration in the PBN_post-LHA-hM4Di mice. (**M**) AAVTransyn-Cre was injected in the LHA and DIO-eNPHR3.0-YFP/DIO-GFP in the RVM of wild-type mice to label the RVM_post-LHA neurons. Optic fiber cannulae were implanted over the RVM at the same coordinates to deliver yellow light. The summary of the fiber placements are graphically represented (right corner). (**N**) YFP-positive cells expressing eNpHR3.0 in the RVM_post-LHA (green) neurons in the RVM. (**O**) Tail-flick latency (seconds) (2.48 ± 0.04 compared to 3.00 ± 0.03, respectively; t-test, ***p=0.0001, n=7) with (ON) and without (OFF) yellow light illumination in RVM_post-LHA-eNPHR3.0-YFP mice, with no significant difference seen in RVM_post-LHA-DIO-GFP mice. (**P**) Number of licks (11.14 ± 1.01 compared to 5.00 ± 0.49, respectively; t-test, ***p=0.0001, n=7), and latency to lick (seconds) (12.71 ± 0.52 compared to 22.86 ± 1.18, respectively; t-test, ***p=0.0001, n=7) with (ON) and without (OFF) yellow light illumination in RVM_post-LHA-eNPHR3.0-YFP mice, with no significant difference seen in RVM_post-LHA-GFP mice.

The online version of this article includes the following figure supplement(s) for figure 8:

**Figure supplement 1.** A poly-synaptic circuit mechanism for dLS mediated SIA.

## Discussion

The dLS has been traditionally considered a key brain region for mediating stress responses (*Anthony et al., 2014*; *Azevedo et al., 2020*; *Kubo et al., 2002*; *Singewald et al., 2011*). However, the role of the dLS in SIA is poorly understood. In this study, we have delineated the downstream partners of dLS and elucidated the mechanisms through which dLS neurons translate stress into pain suppression. We found that the dLS neurons are specifically geared towards coping with stress and, through their connections with the spinal cord via the LHA and the RVM, inhibit responses to noxious stimuli. In conclusion, our study has comprehensively evaluated the involvement of a key neural circuit in SIA by anatomically tracing their connections, monitoring, and manipulating neural activity.

We wondered what the etiologically relevant function of SIA might be, evolutionarily what benefits SIA might provide, and how dLS plays a role in it. SIA enables animals and humans to physiologically and behaviorally evade or cope with stressors in their immediate environment. When the perceived pain is attenuated, attention is drawn toward escaping the stressor. Several experiments support this view, including studies where rats were subjected to mild electric shocks from a floor plate in either an inescapable or an escapable chamber. It was observed that analgesia occurred when the rats could not escape the chamber after experiencing the electric shocks as opposed to when they could escape (*Maier et al., 1982*; *Terman and Liebeskind, 1986*). In line with these studies, we experimentally observed SIA only when mice were restrained for enough time (~1 hr) or when dLS neurons were activated for ~30 min (*Figure 1—figure supplement 2C, D*). We propose that acute restraint drives animals to attempt to escape, and during these attempts dLS neurons are engaged. When the dLS neurons are activated repeatedly, it provides a short-term analgesia.

SIA can be particularly pertinent in individuals with chronic pain, where pathological pain can impede the ability to react to a stressor on time. The role of dLS neurons in chronic pain is unclear. Interestingly, a recent study has shown that the dLS neurons promote both pain and anxiety (*Wang et al., 2023*). This is contrary to our data which shows that dLS activation has analgesic effects. The contradiction may be since dLS neurons may play opposing roles under acute and chronic stress conditions. In contrast to acute stress, mechanisms of which we have explored here, chronic stress which *Wang et al., 2023* studied, is known to exacerbate pain. In addition, the targeted coordinates for dLS used in the study are medial compared to the ones used in our study. However, as mentioned before, lesion studies in humans or animals have consistently indicated that dLS or medial septum stimulation is anti-nociceptive, irrespective of the nature of the noxious stimulus. Notably, chronic stress can cause hyperalgesia, and the underlying circuit mechanisms are poorly understood. It will be interesting to test if the dLS-LHA circuitry is involved in the reduced pain thresholds observed in mice with chronic stress (*Jennings et al., 2014*).

Early studies showed that SIA could either be opioid-dependent or independent (*Watkins and Mayer, 1986*). Naltrexone administration blocked SIA in rats who had undergone electric shocks (*Drugan et al., 1981*; *Maier et al., 1980*). Surprisingly, it was found that acute stress can sequentially induce both opioid dependent as well as opioid-independent SIA (*Grau et al., 1981*). Mutant mice without functional ß-endorphin (endogenous ligand for the mu-opioid receptor, OPRM1) lacked opioid-dependent SIA (mild swim stress), however, they displayed opioid-independent SIA (cold-swim stress) (*Rubinstein et al., 1996*). Interestingly, opioid-dependent SIA was primarily induced when the animals underwent stress in inescapable chambers. Thus, successful induction of SIA may depend on the mode of stress delivery, and the opioid dependence may be decided by the exposure time. Our data also suggests that dLS-mediated RS-induced analgesia is opioid dependent, as we found that SIA induced by dLS activation is reversible by naltrexone administration (*Figure 6—figure supplement 1A–C*). Furthermore, we show that RVM-ON cells play a critical role in dLS-mediated SIA. This observation is supported by a recent study that demonstrated the necessity of kappa-opioid receptors expressing RVM neurons in SIA (*Nguyen et al., 2022*). Together, the restraint-induced silencing of LHA excitatory neurons by dLS may turn downstream mu-opioid receptor-expressing RVM-ON cells amenable to enkephalin and endorphin-mediated modulation and consequent analgesia. Typically, RVM-ON cells target enkephalinergic and GABAergic spinal interneurons in the superficial layers of the dorsal horn that gate pain in turn through their connections with the somatosensory primary inputs to modulate pain (*François et al., 2017*). However, the neurons of our interest in the RVM target the deeper layers (V/VI) in the dorsal horn and may modulate nociceptive thresholds through independent mechanisms (*Figure 6F*). In addition to the opioidergic system, SIA is known to be mediated by the

endogenous endocannabinoid system (*Hohmann et al., 2005*). Brief and continuous electric shocks result in SIA that is not reversed by naltrexone, but can be blocked by endocannabinoid receptor antagonists. Since, the dLS-mediated SIA is suppressed by naltrexone, it is unlikely to be dependent on the endogenous endocannabinoid pathways.

Interestingly, LHA is known to have direct spinal projections and thus can directly modulate pain by passing RVM (*Hancock, 1976*; *Willis and Coggeshall, 2012*). LHA neurons can modulate pain through direct orexinergic inputs to the spinal cord (*van den Pol, 1999*; *Wang et al., 2018*). The direct orexinergic LHA inputs to the spinal cord terminate in the superficial layers of the spinal cord. At the same time, the LHA neurons of our interest provide indirect inputs via RVM to the deeper layers of the spinal cord. Thus, two independent neural pathways (direct and indirect) originating from the LHA may mediate SIA. The orexinergic system housed in the LHA is recruited by acute stress and thus can supplement the LS-LHA-RVM-spinal cord circuitry in stress-induced pain modulation. We did not find direct inputs of LHA$_{post-dLS}$ neurons in the spinal cord with an AAV-mediated labeling strategy; however, these neurons can have partial overlap with the orexin population, and our method may not be sensitive enough to label the axon terminals in the spinal cord.

Notably, LHA neurons are known to respond to stress stimuli (*Owens-French et al., 2022*; *Wang et al., 2021*) and thus can cause SIA independently of dLS. The dLS-independent SIA mechanisms might be driven by the direct orexinergic inputs to the spinal cord from LHA or the PV-expressing LHA neurons can mediate SIA through projections to the PAG (*Siemian et al., 2021*). This is reflected in our results where we observed transient activation of LHA$_{post-dLS}$ neurons when the mice struggled in the RS assay (*Figure 5D*). In addition to the excitatory projections tested here, there are inhibitory neurons in the LHA that project to the PBN (*Moga et al., 1990*). We primarily focused on the excitatory targets of dLS in the LHA, however, LHA is rich in GABAergic neurons and the potential roles of inhibitory targets of dLS in stress-responses remain to be tested. How both excitatory and inhibitory outputs from the LHA can modulate responses to noxious stimuli remains to be investigated. Decades of circuit tracing and functional anatomy studies have revealed synaptic targets of LHA across the brain, including the dLS (*Cassidy et al., 2019*). Such bidirectional connections also exist between the RVM and LHA and exploring the roles of recurrent connections between dLS-LHA/ LHA-RVM in stress-induced pain modulation will further delineate circuit mechanisms of SIA. We developed the microparticle-based CNO administration tool for projection-specific chronic DREADD ligand delivery. The same tool, by tweaking the release rates of the CNO, can be effectively used for chronic neuronal activation/silencing. Finally, the mechanistic interrogation of dLS-centric SIA circuits has revealed a hypothalamic coordinate that effectively connects dLS with RVM to influence nociceptive thresholds. We have primarily focused on thermal nociceptive thresholds, however, it will be interesting to explore how dLS-LHA circuitry influences the effects of stress on other somatosensory modalities, such as itch.

# Materials and methods

## Animals

Animal care and experimental procedures were performed following protocols approved by the IAEC at the Indian Institute of Science. The ethics approval number is CAF/ETHICS/988/2023. The animals were housed at the IISc Central Animal Facility under standard animal housing conditions: 12 hr light/dark cycle from 7:00 am to 7:00 pm with ad libitum access to food and water, mice were housed in IVC cages in Specific pathogen-free (SPF) clean air rooms. Mice strains used: Vglut2-Cre or Vglut2-ires-Cre or *Slc17a6tm2*(Cre) Lowl/J(Stock number 016963); Ai14 (B6;129S6-Gt(ROSA)26Sortm9(CAG-tdTomato)Hze)/J (Stock No 007905), BALB/cJ (Jackson Laboratories, USA). Experimental animals were between 2–4 months old.

## Methods

### Viral vectors and stereotaxic injections

Mice were anesthetized with 2% isoflurane/oxygen before and during the surgery. Craniotomy was performed at the marked point using a hand-held micro-drill (RWD, China). A Hamilton syringe (10 µL) with a glass pulled needle was used to infuse 300 nL of viral particles (1:1 in saline) at 100 nL/min. The following coordinates were used to introduce virus/dyes: dLS- Anterior-Posterior (AP): +0.50, Medial-Lateral (ML): +0.25; Dorsal-Ventral (DV): –2.50; LHA- AP: –1.70, ML: ±1.00; DV: –5.15; RVM- AP: –5.80,

ML: +0.25; DV: –5.25; LPBN- AP: –5.34, ML: ±1.00, DV: –3.15. Vectors used and sources: ssAAV-9/2-hGAD1-chl-icre-SV40p(A) (University of Zurich, Catalog# v197-9), pAAV5-hsyn-DIO-EGFP (Addgene, Catalog# 50457-AAV 1), pAAV5-FLEX-tdTomato (Addgene, Catalog# 28306-PHP.S), pENN.AAV5.hSyn.TurboRFP.WPRE.RBG (Addgene, Catalog# 10552-AAV1), pAAV5-hsyn-DIO-hM3D(Gq)-mCherry (Addgene, Catalog# v141469), pAAV5-hsyn-DIO-hM4D(Gl)-mCherry (Addgene, Catalog# 44362-AAV5), AAV9.syn.flex.GcaMP6s (Addgene, Catalog# pNM V3872TI-R(7.5)), pAAV-Ef1a-DIO-eNPHR 3.0-EYFP (Addgene, Catalog# v32533), pAAV-EF1a-double floxed-hChR2(H134R)-GFP-WPRE-HGHpA(Addgene, Catalog# v64219), AAV1-hSyn-Cre.WPRE.hGH (Addgene, Catalog# v126225), AAVretro-pmSyn1-EBFP-Cre (Donated by Ariel Levine, NIH), AAV retro-hSynapsin-Flpo (Donated by Ariel Levine, NIH), scAAV-1/2-hSyn1-FLPO-SV40p(A) (University of Zurich, Catalog# v59-1), ssAAV-1/2-shortCAG-(pre)mGRASP-WPRE-SV40p(A) (University of Zurich, Catalog# v653-1), ssAAV-1/2-fDIO-(post)mGRASP_2 A_tdTomato(University of Zurich, Catalog# v651-1), pAAV-Ef1a-DIO-tdTomato (Addgene, Catalog# 28306-PHP.S), pAAV-hSyn-fDIO-hM3D(Gq)-mCherry-WPREpA (Addgene, Catalog# 154868-AAVrg), ssAAV-9/2-hSyn1-chl-dlox-EGFP_2 A_FLAG_TeTxLC(rev)-dFRT-WPRE-hGHp(A) (University of Zurich, Catalog# v322-9), AAVretro-hSyn-NLS-mCherry (Donated by Ariel Levine, NIH), AAV9-DIO-GephyrinTagRFP (Donated by Mark Hoon, NIH), AAV9-DIO-PSD95-TagRFP (Donated by Mark Hoon, NIH), AAV5-hSyn-DIO-mSyp1_EGFP(University of Zurich, Catalog# v484-9). For rabies tracing experiments, rAAV5-EF1α-DIO-oRVG (BrainVTA, Catalog# PT-0023) and rAAV5-EF1α-DIO-EGFP-T2A-TVA (BrainVTA, Catalog# PT-0062) were injected first, followed by RV-EnvA-Delta G-dsRed (BrainVTA, Catalog# R01002) after 2 wk. Tissue was harvested after 1 wk of rabies injection for histochemical analysis. Post-hoc histological examination of each injected mouse was used to confirm that viral-mediated expression was restricted to target nuclei.

## Optogenetic and photometry fiber implantation

Fiber optic cannula from RWD, China; Ø1.25 mm Ceramic Ferrule, 200 µm Core, 0.22 NA, L=5 mm were implanted at AP: 0.50, ML: +0.25; DV: –2.50 in the dLS and L=7 mm fibers were implanted at AP: –1.70, ML: ±1.00; DV: –5.15 in the LHA, and AP: –5.80, ML: +0.25; DV: –5.25 in the RVM after AAV carrying GCaMP6s, Channelrhodopsin2 or Halorhodopsin were infused. Animals were allowed to recover for at least 3 wk before performing behavioral tests. Successful labeling and fiber implantation were confirmed post hoc by staining for GFP/mCherry for viral expression and injury caused by the fiber for implantation. Animals with viral-mediated gene expression at the intended locations and fiber implantations, as observed in post hoc tests, were only included.

## Behavioral assays

Behavioral assays for the same cohorts were handled by a single experimenter. Prior to experimentation, the experimenter was blinded to the identity of animals. The genotypes ana/or the mice with AAV1 injections were randomized within cohorts wherever possible. Mice were habituated in their home cages for at least 30 min in the behavior room before experiments. An equal male-to-female ratio was maintained in every experimental cohort and condition unless otherwise stated, with no significant differences seen between sexes in the responses recorded from the behavioral experiments. Wherever possible, efforts were made to keep the usage of animals to a minimum. Deschloroclozapine (DCZ) was diluted in saline (final concentration: 0.1 mg/kg) and injected intraperitoneal (i.p.) 15–20 min before behavioral experiments or histochemical analysis. Mice were injected with intra-plantar (i.pl.) Complete Freund's adjuvant (CFA) 1 d before the behavioral experiments to cause persistent inflammatory pain and thermal hypersensitivity. All the experiments were videotaped simultaneously with three wired cameras (Logitech, USA) placed horizontally and scored offline post hoc manually. The programmable hot plate with gradient function and tail flick analgesiometer (Orchid Scientific, India) were used according to the manufacturer's instructions. For optogenetic stimulations fiber-coupled laser (channelrhodopsin activation; RWD, China), and fiber-coupled LEDs (for halorhodopsin stimulation; Prizmatix, Israel) were used. Prior to the behavioral testing the optic fibers were connected to the cannulae implanted in the mice brain. The animals were habituated for 30 min prior to the commencement of the experiments. The light-dark box tracking and estimations of the time-spent in either chamber were carried out with DeepLabCut and the data were trained and analyzed on a custom-built computer system with AMD Ryzen 9 5900x12 core processor 24 with NVIDIA Corporation Graphics Processing Unit (GPU) (*Mathis et al., 2018*), (*Nath et al., 2019*).

The RS assay was used to induce stress in the experiments reported here. In short, mice were restrained for 1 hr in a ventilated falcon tube, followed by testing them for stress-related and noxious behaviors using the Light Dark Box Assay, the Hot Plate Assay, and the Tail Flick Assay.

Blood corticosterone levels were measured using the Mouse corticosterone Enzyme-linked immunosorbent assay (ELISA) kit (BIOLABS, USA) by collecting blood from wild-type mice, wild-type mice subjected to RS, and dLSGad1-hM3Dq mice administered with DCZ. Plasma extracted from each of the collected blood samples was subjected to Competitive ELISA. Wells pre-coated with corticosterone antigen were incubated with the respective sample in triplicates. Biotin-conjugated primary antibody was added, followed by incubation with streptavidin-horseradish peroxidase (HRP). Finally, the samples turned from yellow to blue in color depending on the concentration of corticosterone present in the sample when Tetramethylbenzidine (TMB) substrate was added. The optical density (O.D.)of the samples was recorded using a spectrophotometer.

The mice were subjected to four primary behavioral assays for the fiber photometry experiments. In the immobilization experiments, the experimenter physically restrained the mice by pressing them down by hand for approximately 10 s. In the tail-hanging experiments, the mice were suspended upside down by their tail for 10 s. In the RS assay, photometry signals were recorded through the fiber-coupled cannulae that passed through a modified RS-inducing falcon tube to allow unrestricted recording. On the hot plate test, the mice were acclimatized to the equipment with the optic fiber connected to the cannulae a day before experimentations. During the experiments, the equipment was first allowed to reach the desired temperature and then the animals were introduced on the hot-plate test.

The collected photometry data was minimally processed with autofluorescence (405 nm) background subtraction and within-trial fluorescence normalization. The median value of data points within the 10 min of home-cage recordings prior to initiation of RS or thermal-plate experiments was used as the normalization factor. Z -score transformation of the fluorescence data was performed with the RWD in-built software.

## Immunostaining, multiplex in situ hybridization, and confocal microscopy

Mice were anesthetized with isoflurane and perfused intracardially with 1 X Phosphate Buffered Saline (PBS) (Takara, Japan) and 4% Paraformaldehyde (PFA) (Ted Pella, Inc, USA), consecutively for immunostaining experiments. Fresh brains were harvested for in situ hybridization experiments. For the cFos experiments, brains were harvested 90 min after RS assay, and 150 mins after i.p. clozapine N-Oxide (CNO) administration. Tissue sections were rinsed in 1 X PBS and incubated in a blocking buffer (2% Bovine Serum Albumin (BSA); 0.3% Triton X-100; PBS) for 1 hr at room temperature. Sections were incubated in primary antibodies in a blocking buffer at room temperature overnight. Antibodies used: cFos: Phospho-c-Fos (Ser32) (D82C12) XP Rabbit mAb #5348, CST; RFP: 600-401-379, Rockland. Sections were rinsed 1–2 times with PBS and incubated for 2 hr in Alexa Fluor conjugated goat anti-rabbit/ chicken or donkey anti-goat/rabbit secondary antibodies (Invitrogen) at room temperature, washed in PBS, and mounted in VectaMount permanent mounting media (Vector Laboratories Inc) onto charged glass slides (Globe Scientific Inc). Multiplex in-situ hybridization (ISH) was done with a manual RNAscope assay (Advanced Cell Diagnostics, USA). Probes were ordered from the ACD online catalogue. We used an upright fluorescence microscope (Khush, Bengaluru) (2.5 X, 4 X, and 10 X lenses) and ImageJ/FIJI image processing software to image, and process images for the verification of anatomical location of cannulae implants. For the anatomical studies,the images were collected with 10 X and 20 X objectives on a laser scanning confocal system (Leica SP8 Falcon, Germany) and processed using the Leica image analysis suite. For the Airy Scan Imaging, Zeiss 980 was used (NCBS Central Imaging Core facility).

## CUBIC clearing and imaging

In order to visualize the fluorescent neuronal labeling in the cleared brain tissue, 300 µm thick sections were cleared by first washing with 1 X PBS for 30 min, followed by 2- hr incubation in 50% Cubic L (TCI, Japan) solution (*Susaki et al., 2020*). Next, the sections were immersed and incubated in 100% Cubic L solution overnight at 37 °C. The sections were preserved and imaged in the Cubic R+ (TCI,

Japan) solution. For imaging the cleared sections, 10 X and 20 X objectives were used along with the Leica SP8 Confocal microscope.

## CNO encapsulation

CNO was encapsulated within poly-lactic-*co*-glycolic acid (PLGA, $M_w$ 10–15 kDa, LG 50:50, PolySci-Tech, IN, USA) microparticles using a single emulsion method. Briefly, 100 mg PLGA and 2 mg CNO were dissolved in 1 mL dichloromethane (DCM) and mixed for 10 min. This mixture was homogenized (IKA T18 digital Ultra Turrax) with 10 mL 1% polyvinyl alcohol (PVA) at 12,000 rpm, resulting in an emulsion. This emulsion was added to 100 mL 1% PVA with magnetic stirring to allow DCM to evaporate. After 4 hr, the microparticles were collected by centrifugation (8000 g) and washed thrice with deionized water to remove PVA. The suspension was frozen, followed by lyophilization to obtain the CNO-encapsulated microparticles as powder. For experiments, the powder was weighed and resuspended in 1 X PBS to get a concentration of 0.5 mg/mL.

## Quantification and statistical analysis

All statistical analyses (t-test and one-way ANOVA test) were performed using GraphPad PRISM 8.0.2 software. ns >0.05, ∗p≤0.05, ∗∗p≤0.01, ∗∗∗p≤0.001, ∗∗∗∗p≤0.0005.

## Acknowledgements

We thank Annappa for providing animal care and facilitating behavioral experiments. We thank the Central Animal Facility for supporting animal experiments. We thank the central bioimaging facilities at IISc for confocal microscopy, and at NCBS, Bengaluru for Airy scan imaging. We thank the DST-FIST program for funding the animal behavioral facility at the Center for Neuroscience.

## Additional information

### Funding

| Funder | Grant reference number | Author |
| --- | --- | --- |
| Wellcome Trust DBT India Alliance | IA/I/19/2/504640 | Arnab Barik |
| Science and Engineering Research Board | CRG/2021/1005124 | Arnab Barik |

The funders had no role in study design, data collection and interpretation, or the decision to submit the work for publication.

### Author contributions

Devanshi Piyush Shah, Conceptualization, Investigation, Methodology, Writing – original draft; Pallavi Raj Sharma, Rachit Agarwal, Methodology; Arnab Barik, Conceptualization, Supervision, Funding acquisition, Project administration, Writing – review and editing

### Author ORCIDs

Pallavi Raj Sharma ⓘ https://orcid.org/0000-0003-3877-7272
Arnab Barik ⓘ https://orcid.org/0000-0001-6850-0894

### Ethics

Animal care and experimental procedures were performed following protocols approved bythe IAEC at the Indian Institute of Science. The ethics approval number isCAF/ETHICS/988/2023.

Reviewer #1 (Public review): https://doi.org/10.7554/eLife.96724.3.sa1
Reviewer #2 (Public review): https://doi.org/10.7554/eLife.96724.3.sa2
Author response https://doi.org/10.7554/eLife.96724.3.sa3

# Additional files

## Supplementary files
MDAR checklist

## Data availability
The raw data for behavior and calcium imaging can be accessed at: https://doi.org/10.5061/dryad.2v6wwpzz9.

The following dataset was generated:

| Author(s) | Year | Dataset title | Dataset URL | Database and Identifier |
|---|---|---|---|---|
| Barik A, Shah DP, Sharma PR, Agarwal R | 2024 | Anatomical, behavioral, and imaging data for a study on neural circuits for stress modulation of pain | https://doi.org/10.5061/dryad.2v6wwpzz9 | Dryad Digital Repository, 10.5061/dryad.2v6wwpzz9 |

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
