## [Editor Report · eLife Assessment]

This **important** work explores the modulation of pain by intense stress. The authors employed a series of cutting-edge techniques and provided **convincing** evidence suggesting that the dorsal lateral septum-> lateral hypothalamus-> rostral ventromedial medulla circuit is responsible for mediating stress-induced analgesia. This work will be of interest to neuroscientists interested in the neural circuits of behavior, and scientists interested in stress or pain.

---

## [Referee Report · Reviewer #1 (Public review)]

The manuscript entitled "A septo-hypothalamic-medullary circuit directs stress-induced analgesia" by Shah et al., showed that the dLS-to-LHA circuit is sufficient and necessary for stress-induced analgesia (SIA), which is mediated by the rostral ventromedial medulla (RVM) in a opioid-dependent manner. This study is interesting and important and the conclusions are largely supported by the data. I have a few concerns as follows:

(1) The present data show that activation of dLS neurons produces SIA, however, this manipulation is non-specific. It may be better to see the effect of specific manipulation of stress-activated c-Fos positive neurons in the dLS using combination of the Tet-Off system and chemogenetic/optogenetic tools.

(2) Depending on its duration, and intensity, stress can exert potent and bidirectional modulatory effects on pain, either reducing pain (SIA) or exacerbating it (stress-induced hyperalgesia,SIH). Whether this circuit in the manuscript is involved in SIH.

(3) It are well-accepted that opioid and cannabinoid receptors participate in the SIA, especially, a critical role of the RVM endocannabinoid system in the SIA, why author focus their study on opioid system?

(4) Whether silencing of the dLS neurons affects stress-induced anxiety-like behaviors? Or， what is the relationship between of SIA and level of stress-induced anxiety?

(5) Please provide the direct electrophysiological evidence for confirming the efficacy of the MP-CNO.

(6) Whether LHA is a specific downstream target for SIA, whether LHA is involved in stress-induced anxiety-like behaviors?

(7) Whether LHA neurons have direct projections to the RVM? If yes, what is its role in the SIA?

---

## [Referee Report · Reviewer #2 (Public review)]

Shah et al. investigate the role of an understudied neural circuitry, specifically the dLS -> LHA -> RVM pathway, in mediating stress-induced analgesia. The authors use a combination of advanced techniques to provide convincing evidence for the involvement of this circuit in modulating pain under stress.

The study begins by mapping the neural circuitry through a series of intersectional tracings. Following this, the authors use behavioral tests along with optogenetic and chemogenetic manipulations to confirm the pathway's role in promoting analgesia. Additionally, fiber photometry is employed to monitor the activity of each brain region in response to stress and pain.

While the study is comprehensive and the findings are convincing, a key concern arises regarding the overarching hypothesis that restraint-induced stress promotes analgesia. A more straightforward interpretation could be that intense struggling, rather than stress itself, might drive the observed analgesic responses.

---

## [Author Response]

**Reviewer #1 (Public Review):**
The manuscript entitled "A septo-hypothalamic-medullary circuit directs stress-induced analgesia" by Shah et al., showed that the dLS-to-LHA circuit is sufficient and necessary for stress-induced analgesia (SIA), which is mediated by the rostral ventromedial medulla (RVM) in a opioid-dependent manner. This study is interesting and important and the conclusions are largely supported by the data. I have a few concerns as follows:

We thank the reviewer for finding our study “interesting”, “important”, and “conclusions are largely supported by data”.

(1) The present data show that activation of dLS neurons produces SIA, however, this manipulation is non-specific. It may be better to see the effect of specific manipulation of stress-activated c-Fos positive neurons in the dLS using a combination of the Tet-Off system and chemogenetic/optogenetic tools.

We agree with the reviewer that activating the stress-“trapped” neurons will be more specific way to induce SIA through septal activation, compared to the activation of entire dLS strategy pursued by us. In most likelihood, we expect to see a robust SIA if specifically stress responsive dLS neurons are observed. We are in the process of acquiring the genetic tools required for “Trapping” stress neurons and expect to be able to perform the experiments suggested by the reviewers in the coming months.

(2) Depending on its duration, and intensity, stress can exert potent and bidirectional modulatory effects on pain, either reducing pain (SIA) or exacerbating it (stress-induced hyperalgesia, SIH). Is the circuit in the manuscript involved in SIH?

As mentioned by the reviewer, it would be reasonable to suspect that the dLS neurons are involved in SIH. However, we believe that the experiments to test this hypothesis is outside the scope of this paper, since here we have focused on the circuit mechanisms for SIA. However, in the revised discussion section, we have included the possibility of dLS neurons driving SIH.

(3) It is well-accepted that opioid and cannabinoid receptors participate in the SIA, and the evidence is especially strong for the RVM endocannabinoid system. Given this, why did the authors focus their study on the opioid system?

We agree with the reviewer that dLS-mediated SIA may work through neural circuits centered on RVM expressing receptors for either or both opioids and endocannabinoids. We primarily focused on the opioidergic system in the RVM as decades of mechanistic work has revealed how the ON, OFF, and neutral neurons modulate pain through the endogenous opioids and even mediate SIA. In the revised discussion, we have included the possibility of involvement of both pain modulatory systems.

(4) Does silencing of the dLS neurons affect stress-induced anxiety-like behaviors? Alternatively, what is the relationship between SIA and the level of stress-induced anxiety?

We did not test if the silencing of dLS would affect stress-induced anxiety, as our focus was on the pain modulatory effects of dLS activation. The relationships between levels of SIA and stress-induced anxiety will be interesting to explore in future. We believe we would need better behavioral assays compared to the existing ones to quantitatively measure levels of stress-induced anxiety and SIA levels.

(5) Direct electrophysiological evidence should be provided to confirm the efficacy of the MP-CNO.

We agree with the reviewer that ex-vivo electrophysiology experiments will substantiate the effectiveness of the MP-CNO. However, we do not have the expertise, or the instrumentation required to perform these experiments in our laboratory.

(6) Is the LHA a specific downstream target for SIA, and is the LHA involved in stressinduced anxiety-like behaviors?

Several lines of evidence points to the fact that LHA neurons are involved in stressinduced anxiety. We have also shown that the dLS downstream neurons in the LHA are activated by acute restraint by fiber photometry recordings. Thus, we expect activation of the LHA neurons will cause stress-induced anxiety. However, we wanted to focus on the pain modulation aspect of the dLS-LHA-RVM circuitry.

(7) Do LHA neurons have direct projections to the RVM? If yes, what is its role in the SIA?

Our anatomical studies using transsynaptic anterograde and retrograde viral strategies in the Figure 6 shows that the LHA neurons have direct projections to the RVM, and these neurons are sufficient in driving hyperalgesia, as well as necessary for SIA.

**Reviewer #2 (Public Review):**
Summary:In this manuscript, Shah et al. explore the function of an understudied neural circuitry from the dLS -> LHA -> RVM in mediating stress-induced analgesia. They initially establish this neural circuitry through a series of intersectional tracings. Subsequently, they conduct behavioral tests, coupled with optogenetic or chemogenetic manipulations, to confirm the involvement of this pathway in promoting analgesia. Additionally, fiber photometry experiments are employed to investigate the activity of each brain region in response to stress and pain.Strengths:Overall, the study is comprehensive, and the findings are compelling.

We appreciate the reviewer for finding our manuscript “comprehensive” and “compelling”.

Weaknesses:One noteworthy concern arises regarding the overarching hypothesis that restrainedinduced stress promotes analgesia. A more direct interpretation suggests that intense struggling, rather than stress per se, activates the dLS -> LHA -> RVM pathway that may drive analgesic responses.

We agree with the reviewer that our data can be interpreted as “intense struggling”, rather than the “acute stress” might have altered the pain thresholds in mice. However, we would like to point out that the restraint induced stress model that we have used has been long regarded as a standard for inducing stress. Moreover, we have demonstrated that dLS activation results into acute stress by measuring the blood corticosterone levels, and showed that dLS activations caused stress-induced anxiety through lightdark box tests.

**Reviewer #2 (Recommendations For The Authors):**
Please find below my other comments for improvements.Introduction: The authors claimed that "dLS neurons receive nociceptive inputs from the thalamus and somatosensory cortices." However, citations are missing.

We have added the citations.

Figure 1 B&C: Although this paper focuses on the dLS, it would be informative to also include vLS c-Fos images (maybe in a supplementary figure), given that these data appear to be already acquired. The inclusion of vLS data will provide critical information regarding potential specificity (or lack of) across LS subregions in stress responses.

In the revised manuscript we have added the vLS c-Fos images as suggested by the reviewer.

Figure 1D: Quantification of Vgat vs. Vglut neurons is missing. It is unclear if the Vgat neurons are restricted to small clusters.

We did not add the Vglut vs, Vgat quantification since from both of our experiments and publicly available data from the Allen Brain Atlas show that almost all of the neurons in the LS are gabaergic. We found very rare,0-2 Vglut2 expressing neurons per section in the the LS of the mouse brain.

Figure 1G: The Y-axis label is missing.

We have added the axis in the revised manuscript.

Figure 2: The authors claimed that dLS neurons are preferentially tuned to stress caused by physical restraint. However, it appears that these neurons are specifically tuned to intense struggle behavior (transient) rather than stress (prolonged).

We agree with the reviewer that the SIA observed in mice with dLS activation, can be interpreted as the effect of transient struggle behavior rather than the prolonged stress. However, we would like to point out that the acute restraint for one hour is known to produce prolonged stress, and is backed up by increased blood coticosterone levels and stress-induced anxiety (Fig1-Fig Supplementary 1).

Figure 4: The authors provided compelling evidence that dLS neurons synapse on LHA Vglut2 neurons. However, it is unclear if they exclusively target the Vglut2 neurons or also synapse on LHA Vgat neurons.

We agree with the reviewer that even though the majority of the dLS downstream neurons in the LHA are glutamatergic, as now shown in the Fig. 4D, few neurons do not express Vglut and thus must be Gabaergic.

Figure 5D: It is unclear if the trace represents dLS or LHA calcium signal (in the main text, the authors claimed both).

Now, we have mentioned the neurons on the LHA we have recorded from at the top of Figure 5C, D.

Figure 6 G&H: Presumably, ΔG-Rabies does not transmit across neurons due to the deletion of the glycoprotein (G) gene. Thus, it is unclear why dLS and LHA neurons express mCherry after injecting rabies into RVM.

The aim of the rabies experiment was to test that the cells in the LHA that receive inputs from the dLS are the same ones that send projections downstream to the RVM. To this end, we used a monosynaptic rabies virus that has retrograde properties. Hence, when injected into the RVM, it was taken up by the terminals of the LHA neurons in the RVM and traveled to the cell bodies in the LHA. We injected the AAV1-Transsyn-Cre in the dLS, so only the cells downstream of the dLS in the LHA can express the Credependent glycoprotein (G) gene. Thus, the rabies-mCherry virus infected the LHA neurons downstream of dLS specifically, and jumped a synapse, to label the upstream dLS neurons.

The authors claim that "RVMpost-LHA neurons may modulate nociceptive thresholds through their local synaptic connections within the RVM, recurrent connections with the PAG, or direct interactions with spinal cord neurons." It is unclear what the "local synaptic connections within the RVM" means. It is also unclear whether there is evidence of recurrent connections between the RVM and PAG.

We meant by local connections as intrinsic connections within the RVM, as in some or few of the RVM neurons, post LHA might be interneurons and mediating SIA by modulating the ON or OFF cells. There are some anatomical evidence for the ascending inputs from RVM to the PAG and the we have now included the citation in the mentioned section of the manuscript.